# Action subsampling supports policy compression in large action spaces

**Shuze Liu** [1]*, **Samuel Joseph Gershman**[2,3]

**1** PhD Program in Neuroscience, Harvard University, Cambridge, Massachusetts, United States of America, **2** Department of Psychology and Center for Brain Science, Harvard University, Cambridge, Massachusetts, United States of America, **3** Center for Brains, Minds, and Machines, Massachusetts Institute of Technology, Cambridge, Massachusetts, United States of America

* shuzeliu@fas.harvard.edu

## Abstract

Real-world decision-making often involves navigating large action spaces with state-dependent action values, taxing the limited cognitive resources at our disposal. While previous studies have explored cognitive constraints on generating action consideration sets or refining state-action mappings (policy complexity), their interplay remains under-explored. In this work, we present a resource-rational framework for policy compression that integrates both constraints, offering a unified perspective on decision-making under cognitive limitations. Through simulations, we characterize the suboptimality arising from reduced action consideration sets and reveal the complex interaction between policy complexity and action consideration set size in mitigating this suboptimality. We then use such normative insight to explain empirically observed phenomena in option generation, including the preferential sampling of generally valuable options and increased correlation in responses across contexts under cognitive load. We further validate the framework's predictions through a contextual multi-armed bandit experiment, showing how humans flexibly adapt their action consideration sets and policy complexity to maintain near-optimality in a task-dependent manner. Our study demonstrates the importance of accounting for fine-grained resource constraints in understanding human cognition, and highlights the presence of adaptive metacognitive strategies even in simple tasks.

**Data availability statement:** All data, analysis code, and the experiment are available at https://github.com/LSZ2001/policycompression_actionconsiderationsets.

## Author summary

This study provides insight into how humans navigate the dual problem of deciding which actions to consider, and which actions to perform in particular contexts. It rationalizes previously observed tendencies of humans to sample generally valuable actions, explores the implication of changing either component of the dual problem on the reward yielded, and demonstrates the framework's relevance through a human experiment. The study suggests that even in simple tasks, humans may still spontaneously simplify the original task to reduce their cognitive load.

**Funding:** The work is supported by the Schmidt Sciences Polymath Program (https://www.schmidtsciences.org/schmidt-science-polymaths/; received by S.G.) and the Kempner Institute for the Study of Natural and Artificial Intelligence (https://kempnerinstitute.harvard.edu/; received by both S.L. and S.G.). There are no grant numbers associated with the above sponsors. The funders had no role in study design, data collection and analysis, decision to publish, or preparation of the manuscript.

**Competing interests:** The authors have declared that no competing interests exist.

## Introduction

Traditional laboratory paradigms for studying decision-making often present human participants with a limited set of predefined alternatives—for instance, choosing between two lotteries with varying risks [1], selecting between multiattribute goods [2], or deciding whether to accept or reject a specific offer [3]. In contrast, real-world decision-making typically involves a vast array of possible actions that are not explicitly presented to agents [4–6]. This complexity is compounded by the fact that the value of different actions often depends on the current environmental context, requiring agents to construct a policy that maps states to actions, either by retrieving rewards from memory or simulating outcomes through forward planning [7,8]. Given the inherent limitations of human cognitive resources [9,10], exhaustively identifying all possible actions and crafting a comprehensive policy is neither practical nor feasible. Understanding how humans navigate this challenge of large action spaces is therefore central to explaining our adaptive success.

A potential solution to the large action space problem involves limiting the number of actions considered (i.e., forming an action consideration set; [11–13]) and/or simplifying the mapping between environmental states and policy-assigned actions (i.e., reducing the state-dependency of the policy) to alleviate cognitive load. These two strategies have been studied separately in prior research. Regarding action consideration sets, evidence suggests that humans tend to generate a small number of actions that have high general value across various contexts (i.e., general-value-based action sampling; [7,14–18]), even when these actions may be suboptimal or even detrimental in specific situations [7]. Additional time and effort are then required to evaluate these actions within the particular context [7,8,19], and such evaluation deteriorates under increased cognitive load. For example, participants' responses across different contexts become more correlated—contrary to what would be expected from purely random noise [20]. These findings have inspired a descriptive two-stage model of open-ended decision-making, which integrates state-independent action generation with state-dependent action evaluation (Fig 1A). However, the optimality of relying on general-value-based action generation remains poorly understood. While previous analyses have explored its benefits from a sample-efficiency perspective, these studies primarily focus on the explicit correlation between general and state-dependent action values in task reward structures [7], without addressing the broader cognitive costs of maintaining state-dependent policies.

Similarly, regarding the mapping between states and policy-assigned actions, prior research has framed the problem using rate-distortion theory, a subfield of information theory [21–23]. The resultant policy compression framework enables the identification of optimal policies and their associated rewards under various levels of cognitive capacity constraints, providing an optimal frontier that aligns with observed human behavior [24,25]. However, the normative foundation of policy compression assumes that the entire action space is known to the agent. Consistent with this assumption, existing studies typically employ tasks with small, well-defined action spaces and symmetric reward structures. This design simplifies the choice of action consideration sets, rendering them trivial and excluding their role from subsequent modeling efforts.

In this paper, we hope to better connect the aforementioned resource constraints native to real-world problems—the number of actions considered and the compression of state-action mappings. To achieve this, we extend the policy compression framework by incorporating action consideration sets. While we continue to assume the existence of a ground-truth full action space from which the optimal rate-distortion frontier can be derived, we propose that humans, through meta-reasoning under resource limitations, instead solve a simplified

rate-distortion problem over a reduced action consideration subset. This new assumption introduces a range of implications for suboptimality that we aim to explore.

By developing this computational-level framework, we seek to address several key questions: 1) Under what conditions is preferential sampling of generally valuable actions advantageous? 2) How do constraints on both policy complexity and action consideration sets interact to produce suboptimality? 3) How do humans empirically navigate such dual constraints, and do they do so in a near-optimal fashion? Using a combination of numerical simulations and human experiments, we investigate these questions in detail and demonstrate the relevance of our normative framework for understanding human decision-making in tasks with moderate to large action spaces.

## Results

### Modeling overview

**The policy compression framework.** The human mind must navigate numerous constraints stemming from the cognitive resources at its disposal [10,26], and these resources have been conceptualized at both physiological [27] and computational levels [28–31]. Here, we specifically focus on the influence of channel capacity, which is the maximum information that can be transmitted across a noisy channel [32,33], on decision-making processes (Fig 1B).

The framework we propose is an application of rate-distortion theory to action selection. Rate-distortion theory describes how to construct an optimal channel that minimizes some notion of error (the distortion)—or, in our case, maximizes reward—subject to a constraint on the information transmission rate [34]. The utility of rate-distortion theory lies in its generality: beyond action selection [23–25], it has been applied to various cognitive processes, including visual working memory [35–37], perception [38], intertemporal decision-making [22], economic behavior under imperfect information [39], cognitive abstraction formation [40], and task-switching costs [41]. Another reason for using rate-distortion theory comes from its information-theoretic nature, which has been classically applied to explain the influence of the number of available actions on response times via the Hick-Hyman Law [42,43]. Given this alignment, rate-distortion theory serves as a fitting framework for our exploration of action consideration sets, offering both theoretical insight and empirical grounding.

For a resource-rational agent, we formalize the cognitive cost as the mutual information between states $s \in \mathcal{S}$ and actions $a \in \mathcal{A}$, which we call the *policy complexity*:

$$I^\pi(S;A) = \sum_s P(s) \sum_a \pi(a|s) \log \frac{\pi(a|s)}{P(a)} \tag{1}$$

where $P(s)$ is the state distribution, $\pi(a|s)$ is the policy, a probabilistic mapping from states to actions, and $P(a) = \sum_s P(s)\pi(a|s)$ is the marginal probability of choosing action $a$. Intuitively, high-complexity policies preserve state information (e.g., deterministic mappings from states to actions) whereas low-complexity policies discard state information (e.g., random actions).

If the agent has infinite cognitive resources, it would be optimal to map each state to the most rewarding action under it. However, we assume that policies are subject to a capacity limit $C$, which acts as an upper bound on policy complexity. Consequently, for a given task with state probability distribution $P(s)$ and state-specific action rewards $Q(s,a)$, agents would maximize trial-averaged reward $V^\pi = \sum_s P(s) \sum_a \pi(a|s) Q(s,a)$ under their constraint

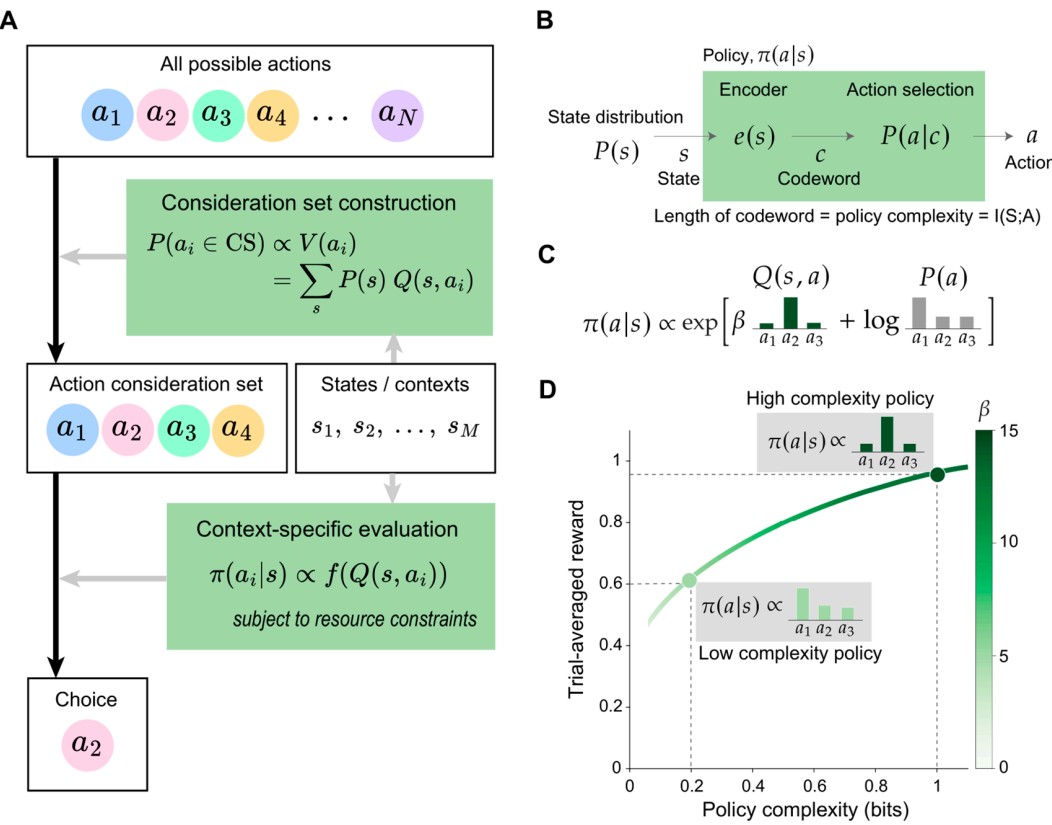

**Fig 1. Theories of action consideration set construction and policy compression. (A)** The two-stage architecture of action consideration set construction and choice [19]. When facing a set of possible actions $\{a_1, ..., a_N\}$, agents first sample an action consideration set, where each action's probability of being sampled is proportional to its general value $V(a_i)$ across all contexts/states $\{s_1, ..., s_M\}$. Then, given the current context $s$, one action is chosen by comparing the context-specific value of all actions retained in the consideration set, $Q(s, a_i)$. **(B)** The policy as a communication channel. A state distribution $P(s)$ generates states $s$ that are encoded into memory via an encoder, $e(s)$, yielding a codeword $c$. The codeword is then mapped onto an action $a$ according to $P(a|c)$. Together, encoding and action selection produce the policy $\pi(a|s)$ that maps states to actions. **(C)** The optimal policy includes a state-dependent term, $Q(s, a)$, and a state-independent term, $\log P(a)$. The $\log P(a)$ term biases choices towards actions that are frequently chosen across all states. The $\beta$ parameter determines the relative contribution of $Q(s, a)$ and $\log P(a)$, controlling the state-dependence of the policy. We highlight distributions for an example state. **(D)** A limit on the channel capacity results in a trade-off between reward and compression. The $\beta$ parameter increases monotonically with policy complexity. We highlight two example optimal policies at different policy complexity levels. The optimal policies trace out the reward-complexity frontier, which delimits achievable performance for a given policy complexity. Panels B-D adapted from [24].

$I^\pi(S; A) \le C$. The above constrained optimization problem prescribes the following optimal policy, which can be empirically found via the Blahut-Arimoto algorithm [44–46]:

$$\pi^*(a|s) \propto \exp[\beta Q(s, a) + \log P^*(a)] \tag{2}$$

where $P^*(a) = \sum_s \pi^*(a|s) P(s)$ is the optimal marginal action distribution, and $\beta$ is a Lagrangian multiplier whose value depends on $C$. Intuitively, at high policy complexity (corresponding to large $C$), the value of $\beta$ is large and the optimal policy is dominated by $Q$-values, which renders it state-dependent. At low policy complexity (small $C$), the value of $\beta$ is close to 0 and $Q$-values have minimal impact on the optimal policy. Moreover,

low-complexity policies are dominated by the $\log P^*(a)$ term, a form of perseveration (state-independent actions; Fig 1C). In general, high-complexity policies enable more trial-averaged reward than low-complexity policies due to their state-dependence. By varying $\beta$ and calculating the optimal policy, we can trace out the reward-complexity frontier (also known as the rate-distortion frontier; Fig 1D), which delimits the maximal trial-averaged reward obtainable for a given policy complexity.

A noteworthy observation is the optimal policy's dependence on the marginal action distribution $P^*(a)$, which grows stronger as policy complexity decreases. In tasks where $P^*(a)$ is uniform over actions, the optimal policy reduces to the traditional softmax choice rule $\pi(a|s) \propto \exp[\beta Q(s, a)]$ [47]. However, in tasks where $P^*(a)$ is non-uniform and biased towards specific actions, then the influence of perseveration is non-trivial. This prediction differs from a traditional softmax model, where the policy approaches a uniform distribution as $\beta$ approaches 0. In recent works, we identified behavioral signatures unique to policy compression—and not predicted by the traditional softmax choice rule—in human data [24,25], which we will replicate in this manuscript.

**Incorporation of action consideration sets.** The above formulation assumes that the agent represents all task information: the marginal state distribution $P(s)$ and the reward structure $Q(s, a)$. However, when the action space is either large or infinite, it becomes infeasible for an agent to consider all possible actions exhaustively and represent them in a policy. Instead, agents may simplify the problem either deliberately or by necessity, considering only a subset of all available actions. This simplification enables agents to address a more manageable subproblem—optimizing and implementing a policy over this action consideration set.

It is straightforward to operationalize action consideration sets in the policy compression framework. Since Eq 2 depends only on the task's $P(s)$ and $Q(s, a)$, one can simply remove the excluded actions from the $Q(s, a)$ matrix, and recompute the optimal policy via the Blahut-Arimoto algorithm. The resultant reward-complexity frontier corresponds to the action consideration set, and one can compare it with the true task's reward-complexity frontier to assess the extent of suboptimality caused by not considering all actions.

We illustrate the proposed analysis on an example task, in which $P(s)$ assigned equal probabilities to all 6 states and $Q(s, a)$ assigns each state a unique optimal action (Fig 2A). We can obtain the reward-complexity frontier, assuming access to the full action space. Given the symmetric nature of $P(s)$ and $Q(s, a)$ in this example task, we can study the influence of partial action consideration sets by removing any number of actions, and similarly deriving reward-complexity frontiers for the remaining action consideration set without loss of generality (Fig 2B). We can hence evaluate the loss in reward associated with using each action consideration set compared to the full-action-space reward-complexity frontier, by subtracting the latter out at each policy complexity level (Fig 2C). This provides a notion of suboptimality along the orthogonal dimensions of policy complexity and action consideration set size.

**Simulation-specific model components.** In simple tasks—such as the one presented in Fig 2 and our later human experiment—it is feasible to derive reward-complexity frontiers for every possible action consideration set. However, this exhaustive approach quickly becomes impractical in more realistic settings with asymmetric reward structures and large action spaces. While an optimal action consideration set exists for any given task and policy complexity level, systematically identifying it is computationally infeasible for agents and difficult to generalize across tasks. This challenge highlights the need for heuristics that guide action

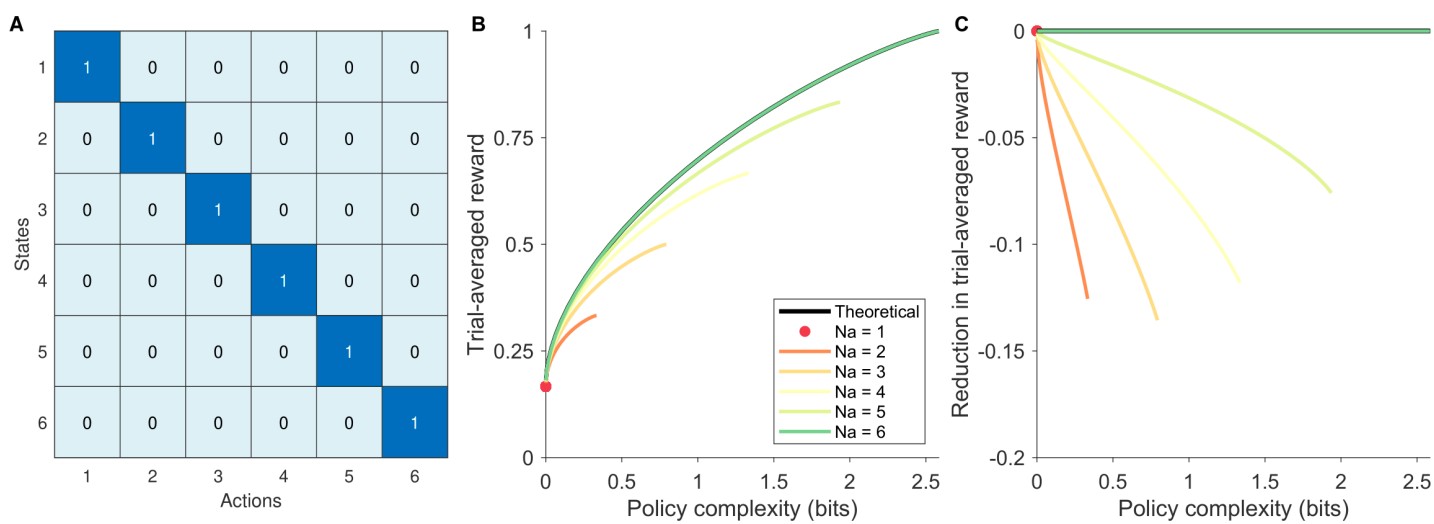

**Fig 2. The influence of partial action consideration sets on a symmetric reward structure task.** (A) The reward matrix $Q(s, a)$ of the example task. Each state has a unique optimal action. We assume that the task has flat $P(s)$ (all states being equiprobable). (B) Reward-complexity frontiers associated with different numbers of actions ($N_a$; colors) retained in the action consideration set. The frontier corresponding to the full action space is colored black. Note that the maximum policy complexity allowed by $N_a$ is $\log_2(N_a)$ bits, such that different colored curves end at different policy complexity levels. (C) The deviation of partial-action-space reward-complexity frontiers to the full-action-space reward-complexity frontier, at different policy complexity levels.

consideration set formation and subsequent policy optimization—heuristics that agents could learn over longer timescales, amortize in memory, and apply flexibly to new tasks [48].

Motivated by the challenge above, we explore the advantages of biased action sampling and the interplay between policy complexity and action consideration sets by employing numerical simulations in more complex task setups. Such simulations require modeling action proposal distributions and incorporating bias-correction algorithms, both of which serve as tractable heuristics for constructing effective action consideration sets and approximating optimal policies.

**Action proposal distributions.** Following the two-stage decision-making architecture described earlier (Fig 1A) [19], we first consider the proposal distribution $P_0(a)$ through which actions are sampled and enter the consideration set. The first two candidates are the flat distribution and the general-value-based distribution, as specified in past works [7]. Given the policy compression framework, a third proposal distribution of interest is the optimal policy's marginal distribution $P^*(a)$ at whichever policy complexity level the agent commits to, assuming the full action space. This proposal distribution is an oracle, because one needs to know the optimal policy to find it. However, it serves as a benchmark for the performance of the previous two candidates. To summarize, we consider three candidate proposal distributions:

1. Flat distribution: $P_0(a) = \text{const.}$
2. General-value-based distribution: $P_0(a) \propto V(a) = \sum_s P(s) Q(s, a)$
3. Oracle distribution: $P^*(a) = \sum_s P(s) \, \pi^*(a|s)$

Another model parameter worth exploring is whether actions are sampled with or without replacement from these proposal distributions. This would impact the bias-correction algorithms we introduce below.

**Bias-correction algorithms.** After sampling actions from the proposal distribution $P_0(a)$, the simplest solution is to perform the Blahut-Arimoto (BA) algorithm on the action sample. The BA algorithm would enable us to find the optimal policy at multiple policy complexity levels, assuming that the action consideration set equals all unique actions sampled. Notably, BA is indifferent to how often each action is sampled—it considers only the number of distinct actions in the sample, without accounting for their frequencies. Furthermore, BA does not correct for any bias in the $P_0(a)$ that may make some actions more likely to be sampled than others.

In contrast to BA, a truly bias-correcting algorithm is self-normalized importance sampling (SNIS), which assumes sampling actions with replacement and is informed by the repetitive sampling of the same action [49,50]. We consider SNIS a compelling alternative algorithm to examine for two main reasons. First, SNIS has previously been shown to account for the human tendency to oversample extreme outcomes when estimating the expected utilities of actions [51]. Second, similar importance sampling algorithms have successfully explained a wide range of human behaviors under resource constraints, including multiple object tracking [52], concept learning [53], reinforcement learning [54], and sentence parsing [55]. While sampling outcomes and actions are technically different domains, we postulated that SNIS may be an equally effective algorithm for action selection.

According to SNIS, we assume that independent action samples $(\alpha_1, \alpha_2, ...\alpha_n)$ are drawn with replacement from the proposal distribution $P_0(a)$. We can then construct the following estimator for $\pi^*$ under the full action space:

$$\hat{\pi}^*(a|s) \propto \sum_{j=1}^{n} \exp\left(\beta Q(s, \alpha_j)\right) \frac{P^*(\alpha_j)}{P_0(\alpha_j)} \mathbb{I}(\alpha_j = a), \tag{3}$$

where $\mathbb{I}(\alpha_j = a)$ is the indicator function comparing $\alpha_j$ to $a$. Intuitively, any bias in the action proposal distribution would reflect in both the counts $\mathbb{I}(\alpha_j = a)$ and the denominator $P_0(a)$, such that they would cancel out in the asymptotic limit of infinite samples and thus achieve bias correction. Through SNIS, $\hat{\pi}^*(a|s)$ is an asymptotically unbiased estimator of $\pi^*(a|s)$ at the same $\beta$ value, and can be used for action selection.

The BA and SNIS algorithms differ in their underlying objectives. BA is designed to maximize trial-averaged reward given a specific action consideration set and varying levels of policy complexity [44,45]. In contrast, SNIS does not aim to optimize rewards within the current consideration set but instead prioritizes unbiasedness—seeking to approximate the full-action-space optimal policy at the same $\beta$ without introducing bias. Later in the paper, we assess how SNIS's objective influences reward attainment. Using an approach similar to BA in Fig 2C, we evaluate the average deviation of $\hat{\pi}^*$ (computed for each action sample $(\alpha_1, ..., \alpha_n)$) from the true task's reward-complexity frontier, across different sample sizes $n$. This deviation quantifies the potential loss in achievable reward.

Note that the SNIS assumption of sampling actions with replacement requires a different resource formulation—not the action consideration set size (i.e. the number of distinct actions considered, $N_a$), but the number of action samples ($n$). There is limited evidence on whether humans sample actions with or without replacement, and hence we will explore both cases. Luckily, the simpler BA algorithm is compatible with both sampling methods—when sampling actions with replacement, it simply does not make use of how often each action is sampled, relying only on the set of unique actions. Hence, we can fairly compare BA and SNIS in the regime of sampling with replacement. To summarize, we consider two bias-correcting algorithms in the simulations:

1. Running Blahut-Arimoto on the retained actions (BA): no bias correction, applicable to sampling actions both with and without replacement.
2. Self-normalized importance sampling (SNIS): has bias correction, applicable only to sampling actions with replacement.

## Exploring conditions that favor general-value-based action sampling

To explore the normative basis of general-value-based action sampling, we need to understand the cognitive constraints that likely define the underlying optimization problem. First, humans frequently operate under low policy complexity. As demonstrated in previous research, most human participants adopt lower policy complexity than what is maximally possible or reward-maximizing under various tasks, as exerting high policy complexity incurs both memory and time costs [24,25]. Second, humans frequently operate over small action consideration sets. There is ample evidence in the option generation literature suggesting that humans generate and consider very few options before committing to a decision [14–18], implying that mechanisms for generating actions must respect this sample size constraint. Based on these observations, we propose that general-value-based sampling serves as a task-general heuristic for action selection, specifically in ecologically relevant regimes of low policy complexity and small action consideration sets. By leveraging this heuristic across different tasks, agents can achieve satisficing outcomes without fully optimizing action consideration sets for each specific task [56].

The above hypothesis is also intuitive under reasoning. First, at low policy complexity, even when given access to the full action space, the optimal policy should still place all its probability mass on the action that has the highest general value, as prescribed by rate-distortion theory [57] given the $P^*(a)$ term in Eq 2. In other words, optimality at near-zero policy complexity depends solely on whether the action with highest general value is sampled, thus making general-value-based action sampling beneficial. However, such promises no longer hold as policy complexity increases, where the optimal policy becomes sensitive to the state-specific reward of actions, which no longer perfectly correlates with their general value [7]. Second, constraints on action consideration set size should exacerbate the above low-complexity advantage, because smaller consideration sets further restrict the probability of sampling this highest-general-value action under a uniform proposal distribution.

We first assess our hypotheses on a large action space task with 16 states under uniform $P(s)$ and 32 actions with random $Q(s, a)$ entries (Fig 3A). Unlike the symmetric task in Fig 2, the current task structure prevents humans from exhaustively enumerating all possible action consideration subsets and determining the one that is closest to optimality (defined as being close to the full-action-space reward-complexity frontier), thus necessitating action sampling heuristics.

Assuming no bias correction (i.e., running BA on the action consideration set, as opposed to SNIS), our predictions hold. The general-value-based proposal distribution provides a clear advantage at low policy complexity and small action consideration set sizes. Compared to the uniform proposal distribution, it yields trial-averaged rewards that are closer to those obtained with the oracle proposal distribution (Fig 3B Row 1). Additionally, its proximity to the $y = 0$ line suggests lower suboptimality, indicating that it achieves reward levels closer to the theoretical maximum for the given policy complexity. However, the conclusion differed when SNIS is used coupled with sampling with replacement. The bias correction has encouraged the resultant policy to deviate away from more frequently sampled actions, hence eliminating the advantage of general-value-based proposal distributions even at low policy complexity (Fig 3B Row 2). In fact, if one compares SNIS and BA, both assuming sampling

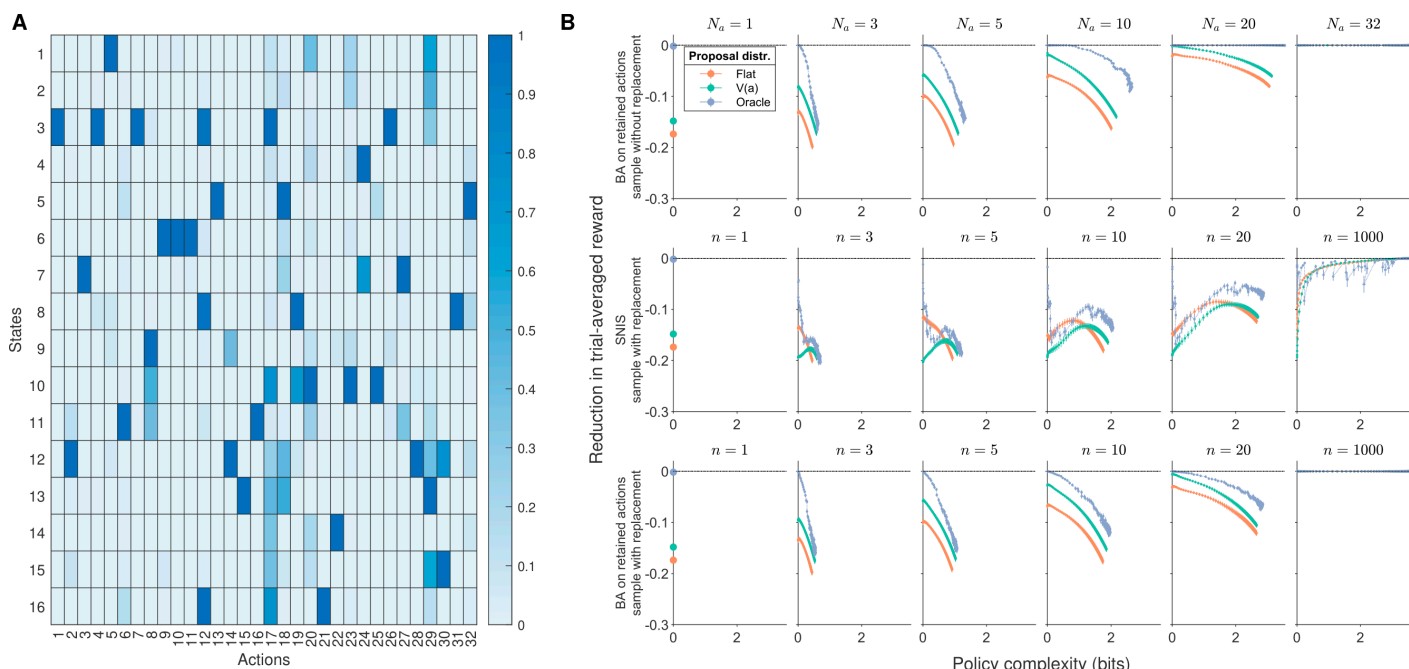

**Fig 3. Simulating the suboptimality of partial action consideration sets, in a random reward structure task.** (A) Reward structure of the task. (B) Simulation results. Colors denote proposal distributions: flat $P_0(a)$ = const., general-value based $P_0(a) \propto V(a)$, or oracle $P_0(a) \propto P^*(a)$ under the full action space at the same $\beta$. Columns denote different action consideration set sizes for sampling without replacement, and different action sample sizes for sampling with replacement. **Row 1:** Assuming running Blahut-Arimoto on the retained action consideration set (BA) and sampling without replacement, the loss in trial-averaged reward from optimality under different proposal distributions (color) and different action sample sizes (subpanels). 2D errorbars denote mean±SEM of policy complexity and reduction in trial-averaged reward over random simulations, aggregating over simulations that share the same $\beta$. Different columns correspond to different action consideration set sizes (i.e., number of distinct actions considered; $N_a$). **Row 2:** Assuming running self-normalized importance sampling (SNIS) and sampling with replacement. Different columns correspond to different action sample sizes ($n$). **Row 3:** Assuming running BA and sampling with replacement.

with replacement, the benefit of BA is evident: policy performance was much closer to optimal at low policy complexity levels, whereas convergence to the optimal policy also occurred under smaller action sample sizes (Fig 3B Row 3). These results suggest that apart from the low policy complexity and small action consideration set regimes, the normative basis of using general-value-based sampling is dependent on the downstream assumption of agents using the best policy under their action consideration set (BA), without significant bias-correction for their underlying proposal distribution (SNIS).

Knowing the limitations of using a single example task, we again performed identical simulations using two more tasks featuring different action general values: a large-action-space task with scarce rewards $Q(s,a)$ concentrated on very few actions (S2 and S3 Figs), and the human experiment task to be described later (S4 and S5 Figs). Generally speaking, using BA over the action consideration set outperformed SNIS in terms of achieving near-optimal policies at various policy complexity levels; and it was under this downstream BA algorithm—and the corresponding assumption of maximizing reward under the retained action consideration set—that general-value-based action sampling yielded advantage at low policy complexity and fewer actions considered. In other words, the hypothesis of general-value-based sampling as an adaptation is not generally true across all downstream algorithms, as shown through the SNIS simulations.

## The interplay of policy complexity and action consideration set size

Observing the suboptimality curves in detail, it becomes evident that their shapes under BA and general-value-based action sampling (either with and without replacement) conforms to intuitions that motivated our hypothesis, as well as analytical results for the simpler task (Fig 2). This allows us to extract general insight on the interaction between the two sources of cognitive constraints—policy complexity and action consideration set size—based on the BA simulation results.

First, larger action consideration sets enable higher maximum policy complexity (which is equal to $\log_2\left(\max(\{|\mathcal{S}|, N_a\})\right)$ bits), until the action consideration set size/number of distinct actions considered $N_a$ exceeds the state set size $|\mathcal{S}|$. Second, across all simulated tasks, the suboptimality in trial-averaged reward increases with policy complexity, whereas enlarging the action consideration set size (or taking more action samples assuming sampling with replacement) mitigates the slope of this suboptimality increment (making it more positive; Figs 2, 3, and S1–S5). Overall, assuming optimal performance over the remaining action consideration set, the interplay between policy complexity and action consideration sets can be summarized as follows:

1. Larger action consideration sets enable higher policy complexity, which, under the reward-complexity-frontier, enables higher trial-averaged reward.
2. Larger action consideration sets mitigate the increase in suboptimality following increases in policy complexity, hence contributing to the reward-efficiency of policy complexity increments.

How human agents utilize these two interplaying properties is subject to further investigation. Depending on their strategies, the above two effects may fail to manifest behaviorally. For example, if humans do not increase policy complexity along with their action consideration set size, the first effect would be nullified. Similarly, if humans cannot efficiently learn, memorize, or implement the optimal policy for larger action consideration sets (likely true given set size effects observed in working memory studies; see [31,58]), the second effect would be weakened. It hence becomes an interesting question to study how humans navigate the policy complexity-action consideration set size landscape, as they deliberately simplify the original task's action space and bear the suboptimal consequences for doing so.

## Human behavior reflects joint constraints on policy complexity and action consideration set size

As much as we want to study human action subsampling in naturalistic, open-ended problems, the challenge is that we, as experimenters, cannot easily determine the relative frequency of states $P(s)$, the full set of possible actions $\mathcal{A}$, and each individual's subjective reward for every state-action pair $Q(s, a)$ for normative analysis. However, the cognitive processes involved—the retention of certain subsets of actions, and the establishment of resource-constrained policies that map states to actions—could be manifested in simpler controlled experiments. We hence focused on a contextual multi-armed bandits task that preserved crucial problem features of interest.

**Experimental task.** We manipulated response time (RT) deadlines consistent with previous works on policy compression and option generation [20,24], while leaving the state distribution $P(s)$ and reward structure $Q(s, a)$ fixed. Participants completed three blocks of trials with RT deadlines 2s, 1s, and 0.5s respectively in random order. On each trial, participants see one of six possible images (state) and must press one of seven keys (actions) within the

RT deadline. Reward delivery was deterministic: each state corresponded to a unique reward-maximizing (optimal) action, and the remaining action is a "safety" action that yielded a small but positive reward regardless of the current state (Fig 4A and 4B). Failure to respond within

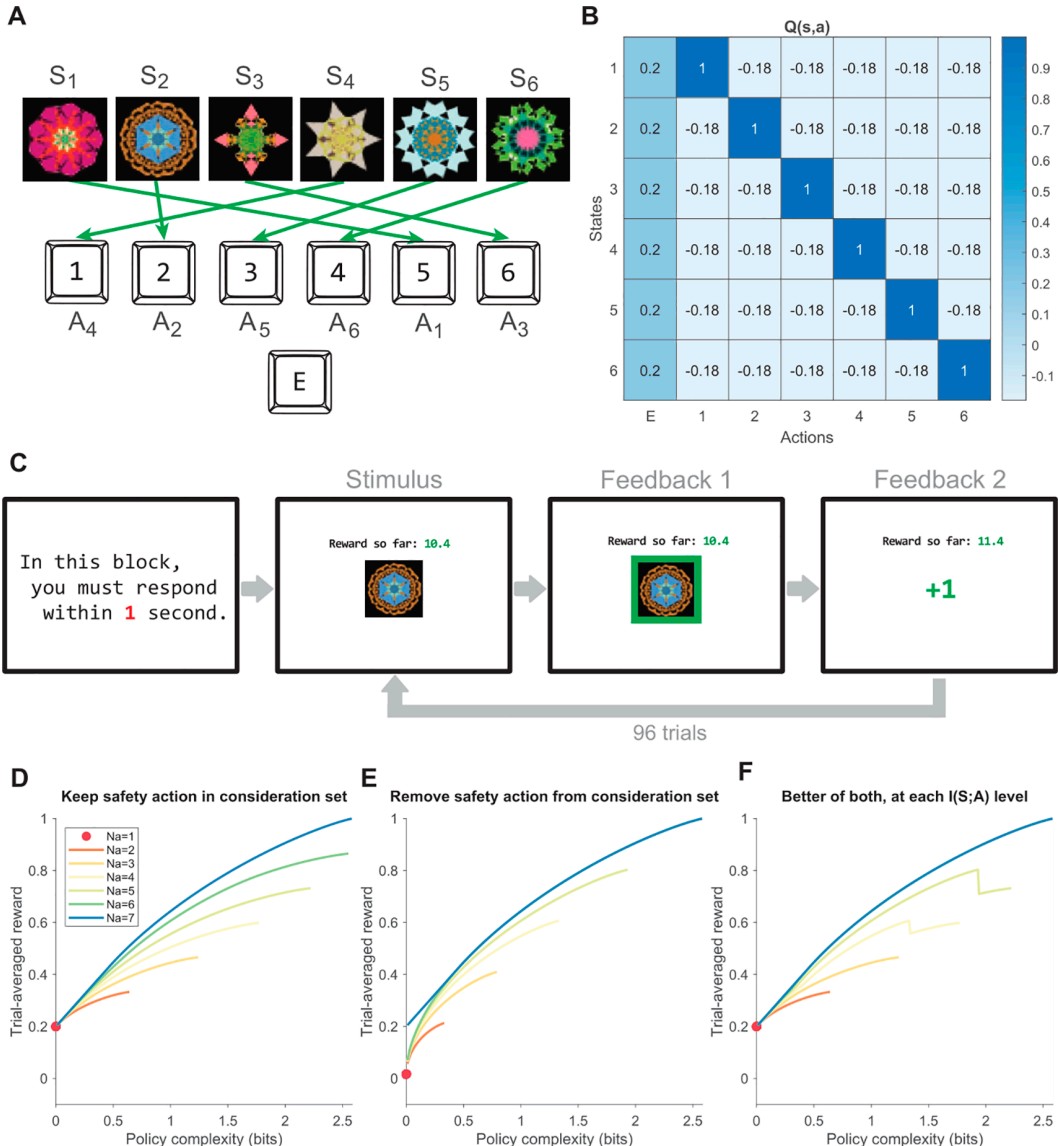

**Fig 4. Human experiment task setup. (A)** The six possible states (images) and the corresponding optimal actions (key presses). The optimal mapping between images and keys was randomized across participants. The example stimuli images visualized are adapted from [59] and differed slightly from those used in the actual experiment. **(B)** The reward structure $Q(s, a)$ of the task. There is a safety action that yields deterministically a low but positive reward for all states. Within each block, we counterbalance the number of trials where each state is used, such that the empirical $P(s)$ is flat. **(C)** On every trial, the participant observes an image (state) and responds by pressing a key (action). Then, reward feedback is provided as a border around the image where the border color denotes reward, and then as a numeric reward value, before the next trial starts. Participants can track cumulative reward (number above image) for the block. After the block ends, participants receive feedback on the total reward gained in the block. Participants are informed of the block's RT deadline before starting. **(D)** Reward-complexity frontiers under different action consideration set sizes, where the safety action is retained in the set. **(E)** Reward-complexity frontiers under different action consideration set sizes, where the safety action is not retained in the set. **(F)** The theoretical reward-complexity frontier under different action consideration set sizes, constructed by finding the maximum between (D) and (E) for each policy complexity level and action consideration set size.

the RT deadline resulted in a −1 reward penalty. After making a response, participants receive feedback on their reward yielded, before being redirected to the next trial (Fig 4C). Due to the simplicity of the task, we can exhaustively enumerate reward-complexity frontiers associated with different action consideration sets, depending on their size ($N_a$) and whether the safety action is included (Fig 4D and 4E). We can then find the higher of the two frontiers associated with the same consideration set size, and visualize it as the $N_a$-specific reward-complexity frontier (Fig 4F). Inspection of these frontiers reveal that the normative interplay between policy compression and action consideration set size, as disclosed by previous simulations, holds in this experimental task, thus enabling our predictions below.

**Experiment predictions.** We make the following predictions regarding policy compression alone, which help assess the behavioral relevance of this normative framework: shorter RT deadlines should be associated with 1) lower policy complexity; 2) lower trial-averaged reward; 3) higher probability of choosing the safety action. The first prediction arises from previous works showing that policy complexity incurs time costs [24,25]; the latter two predictions arise from the task's reward-complexity frontier, which takes into consideration the safety action having higher general value than other actions, and that it should always be chosen at policy complexity 0 bits. Given the time cost implication of policy complexity, we also predict that 4) higher policy complexity incurs longer RTs.

Regarding the interplay of policy complexity and number of distinct actions chosen, we make the following predictions: 5) shorter RT deadlines should be associated with fewer numbers of distinct actions chosen, and 6) higher numbers of distinct actions chosen incur longer RTs, because both predictions are consistent with previous accounts of set size effects on RT as well as sampling costs [30,42,43]. Based on the simulation results, we also predict that 7) In terms of achieving higher trial-averaged reward, policy complexity has a positive effect, while increasing the number of actions chosen exacerbates the above positive effect; 8) In terms of achieving lower suboptimality (loss in trial-averaged reward compared to the full-action-space reward-complexity frontier at the same policy complexity level), policy complexity has a negative effect, while increasing the number of actions chosen mitigates the above negative effect. A normative corollary of the predictions above is that 9) policy complexity and the number of actions chosen should be positively correlated, to mitigate any suboptimalties. Note that while the number of actions chosen places an upper bound on policy complexity, participants still have freedom to empirically adjust one independent from the other across task conditions (e.g., incorporate more actions while maintaining low policy complexity), such that nonsignificant correlations are in principle possible. Hence, any correlations identified in 9) may still be interpreted qualitatively with theoretical significance.

Orthogonal to the policy compression-action consideration set framework, we also made predictions regarding how a human participant decides on their policy complexity level and action consideration set size. We postulate that higher training block accuracy rates (averaged over training blocks) should be associated with 10) higher mean policy complexity and 11) higher mean number of actions chosen across test blocks.

The predictions above are validated using a combination of linear mixed-effects (LME) regression models and Pearson correlation coefficients. The statistical analysis details are elaborated in Methods.

**Experimental results.** Consistent with previous studies [21,24,25], most participants achieved trial-averaged reward levels that were not far from the full-action-space reward-complexity frontier (Fig 5A). However, given the task's large action space, visible suboptimalities did exist. Most participants lay closer to the ranges of their $N_a$-specific reward-complexity

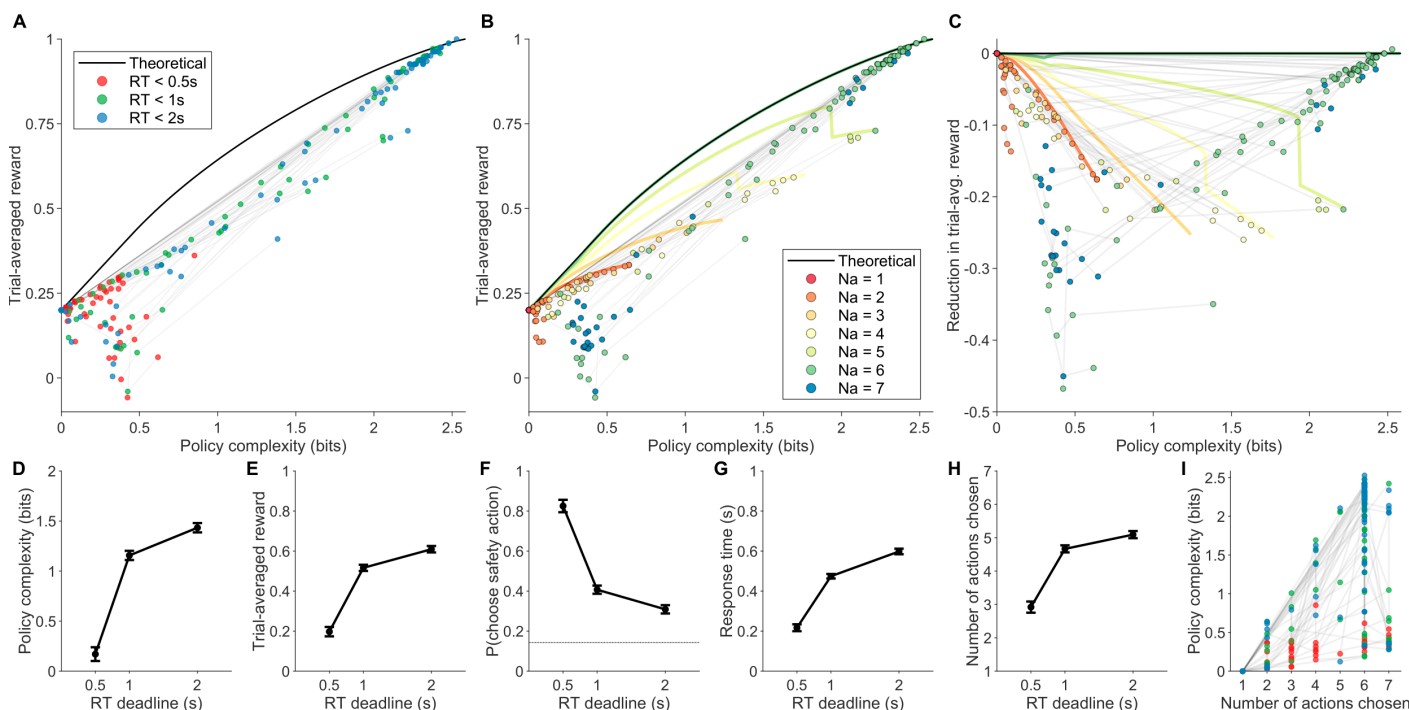

**Fig 5. Human experiment behavioral results. (A)** Trial-averaged reward and policy complexity of participants across RT deadline conditions (colored points), and the full-action-space reward-complexity frontier (black line) at each policy complexity level. Semitransparent gray lines between points connect the same participant's data points in different RT deadline conditions. **(B)** The same figure as (A), but visualizing the number of distinct actions taken (color) by each participant in each condition, and the reward-complexity frontier associated with each number of distinct actions taken (colored lines). **(C)** The same figure as (B), but visualizing each data point's deviation from the full-action-space reward-complexity frontier. **(D-H)** Mean±SEM of participant trial-averaged reward (D), probability of choosing the safe action (E), policy complexity (F), number of distinct actions chosen (G), and mean RT (H) across RT deadline conditions. All SEM errorbars are within-participant [60]. **(I)** Correlating policy complexity and number of distinct actions chosen. The color scheme is identical to (A).

curves, indicating that the number of actions chosen could qualitatively explain some portion of participant suboptimality. However, there also existed a cluster of extremely suboptimal participants with low policy complexity and $N_a$ = 6 or 7, who were earning much lower trial-averaged reward compared to that yielded by always choosing the safety action ($y$ intercept of all reward-complexity curves at $y$ = 0.2; Fig 5B and 5C). In the Supporting Information, we re-performed all analyses after removing this cluster of $N$ = 13 participants (post-hoc; identified based on an empirical trial-averaged reward $\leq$ 0.15 cutoff to accommodate occasional key-pressing errors), thus concluding that our results reported below were not solely the result of significant participant suboptimality (Table A in S1 Appendix).

In line with typical predictions of policy compression, during long RT deadline blocks participants used policies with higher complexity (fixed effects 0.762 $\pm$ 0.0778, $t(223)$ = 9.78, $p < 10^{-18}$, random effects SD = 0.401; Fig 5D), earned higher trial-averaged reward (fixed effects 0.249 $\pm$ 0.0265, $t(223)$ = 9.37, $p < 10^{-17}$, random effects SD = 0.144; Fig 5E), chose the safety action less frequently (fixed effects –0.309 $\pm$ 0.0325, $t(223)$ = –9.50, $p < 10^{-17}$, random effects SD = 0.0910; Fig 5F), and consequently incurred longer RTs (fixed effects 0.236 $\pm$ 0.0203, $t(223)$ = 11.6, $p < 10^{-23}$, random effects SD = 0.112; Fig 5G). More importantly, individual participant RT related positively to their policy complexity levels (fixed effects 0.273 $\pm$ 0.0103, $t(223)$ = 26.3, $p < 10^{-69}$, random effects SD = 0.0468; S6B Fig). These results suggest that the policy compression framework offers a good first-hand description of human behavior in our task, thus motivating further analyses.

We next tested predictions related to the number of actions chosen. As predicted, participants increased their number of actions chosen for longer RT deadline conditions (fixed effects $1.30 \pm 0.159$, $t(223) = 8.19$, $p < 10^{-13}$, random effects $SD = 0.0488$; Fig 5H), featuring adequate variability needed for condition-specific analyses. Also, Individual RT related positively to their number of actions chosen (fixed effects $0.0861 \pm 0.00671$, $t(223) = 12.8$, $p < 10^{-27}$, random effects $SD = 0.0282$; S6C Fig). To assess whether the number of actions chosen contributes to RT in a way independent from its influences on policy complexity, we ran a post-hoc LME analysis and found significant positive effects for both policy complexity (fixed effects $0.375 \pm 0.0510$, $t(221) = 7.35$, $p < 10^{-11}$, random effects $SD = 0.0568$) and the number of actions chosen (fixed effects $0.0146 \pm 0.00625$, $t(221) = 2.34$, $p = 0.0200$, random effects $SD = 0.0177$). However, the statistical significance of the number of actions chosen was lost when we changed the threshold for action counting, reducing it to a trend (see Tables B and C in S1 Appendix). Given these ambivalent results, we postulate that classical empirical results connecting the number of actions to RT may be largely explained by policy complexity, a direction that warrants future testing [42,43].

We next tested the most important framework predictions regarding the interplay between policy complexity and the number of actions chosen. As predicted, in terms of trial-averaged reward, policy complexity contributed positively (fixed effects $0.248 \pm 0.0209$, $t(221) = 11.8$, $p < 10^{-24}$, random effects $SD = 0.0768$), while the interaction term supported our hypothesis that increasing the number of actions chosen would exacerbate the above positive relationship (fixed effects $0.0205 \pm 0.00306$, $t(221) = 6.70$, $p < 10^{-10}$, random effects $SD = 0.00659$). Similarly, in terms of reducing loss in trial-averaged reward compared to optimal, policy complexity contributed negatively (i.e. enlarging the suboptimality; fixed effects $-0.276 \pm 0.0223$, $t(221) = 12.4$, $p < 10^{-26}$, random effects $SD = 0.0864$), while the interaction term supported our hypothesis that increasing the number of actions chosen would mitigate the above negative relationship (fixed effects $0.0571 \pm 0.00341$, $t(221) = 16.7$, $p < 10^{-40}$, random effects $SD = 0.0103$). These results suggest that the normative interaction between policy complexity and the number of actions considered, as revealed by the simulations, reflects strongly in the reward and suboptimality patterns of human behavior in our task.

We then assessed the correlation between policy complexity and the number of actions chosen. As predicted, the Pearson correlation was positive and significant ($R = 0.671$, $p < 10^{-30}$; Fig 5I). This correlation, however, was not strong enough to introduce multicollinearity problems in previous LME regressions (variance inflation factor (VIF) = 1.82), such that their coefficient estimates are still trustworthy. One potential concern is that the number of actions chosen ($N_a$) places an upper bound on policy complexity ($\log_2(N_a)$ bits), which raises the alternative explanation that the observed correlation simply reflects a uniform sampling of policy complexity within its allowable range. To address this, we performed post hoc Kolmogorov-Smirnov tests on the empirical distribution of policy complexity at each $N_a$. These tests revealed significant deviations from uniformity for $N_a = 2, 3, 6, 7$ ($p < 10^{-4}$). Moreover, visual inspection of the policy complexity distributions revealed a shift from right-skewness to left-skewness as $N_a$ increased from 2 to 6 (S7 Fig). This pattern may explain the inability to reject uniformity for intermediate $N_a$ values ($N_a = 4, 5$), and supports the prediction that humans tend to increase both $N_a$ and policy complexity across RT deadline conditions in an effort to reduce suboptimality.

We assessed our final hypothesis that participants set their policy complexity and number of actions chosen in a way related to their training block accuracy. As predicted, training accuracy correlated positively with mean test block policy complexity ($R = 0.792$, $p < 10^{-16}$; S6E Fig) and mean test block number of actions chosen ($R = 0.327$, $p = 0.00416$; S6F Fig). We wondered if the ability to adjust policy complexity and number of actions chosen based

on training accuracy is related to suboptimality, and hence performed additional post-hoc analyses on the $N$ = 13 cluster of suboptimal subjects identified earlier in Fig 5A–5C. Subgroup analyses revealed that these subjects did not significantly adjust their number of actions chosen based on training accuracy (which is mostly low; $R$ = 0.0740, $p$ = 0.810), whereas the remaining $N$ = 62 subjects did ($R$ = 0.907, $p < 10^{-4}$; Table A in S1 Appendix).

## Discussion

In this paper, we explored the relationship between two factors in action selection: policy complexity—the compression of the mappings between states and actions, and the size of action consideration sets—which and how many actions are considered. We investigated the two aforementioned factors by running simulations and analyzing human action selection in tasks with large action spaces. Unlike studies of open-ended problems, our experimental setup ensured that the ground-truth action space and reward structure remain accessible for the computation of optimal policies. This allowed us to develop a framework that jointly considers policy complexity and action consideration sets, and assess its relevance to human behavior.

Our framework provides normative insight on previous findings in the domain of human action/option generation. First, we clarified the empirical observation that humans preferentially generate actions with high general value over environmental states [7,8,19,20], by identifying conditions in which such biased action sampling is beneficial. Across simulations over multiple tasks, the benefit depended on multiple factors: the task structure, policy complexity, the action consideration set size, and how agents derive an approximately optimal policy given their retained action consideration set. However, we found that preferential sampling of generally-valuable actions conferred a robust advantage under low policy complexity, and became increasingly beneficial when the number of actions considered is small. Low policy complexity and small action consideration sets are reasonable resource constraints that humans operate over [24,25,30]. This justifies the usage of general-value-based action sampling as a fast and frugal heuristic readily applicable across tasks [56]. Second, through introducing well-substantiated assumptions relating policy complexity to cognitive load and RT [24,25], we also provided normative justification for the neighboring phenomenon that people's answers become more correlated across question contexts under shorter RT deadlines, instead of being subject to greater independent noise [20]. The framework concludes that this behavior is rational if the answers for all contexts converge to answers with the highest general value, which constitutes the optimal policy at near-zero complexity. The insights yielded from our framework contribute to the study of resource-rationality, which analyzes human cognition in terms of solutions to constrained optimization problems [10,61].

A separate intriguing point is the poor performance of SNIS as a downstream algorithm for policy approximation, despite its guarantee of asymptotic unbiasedness. By correcting biases in a general-value-based action proposal distribution, SNIS inadvertently removed the latter's advantage at low policy complexity, and prolonged convergence to the full action space's optimal policy in multiple task setups. The above finding provides a cautionary example of applying statistical approximation algorithms to understanding human behavior [51,62,63]. Given the resource constraints that humans face, asymptotic optimality may not be enough to motivate the usage of a particular algorithm in naturalistic settings, and a case-by-case assessment of their performance in low sample-complexity regimes may be necessary [30].

Through both simulations and exhaustive enumeration of action consideration sets over simpler tasks, our framework formalizes the relationship between policy complexity and

action consideration set size. The two resource constraints manifest differently in normative analysis: assuming near-optimal behavior under the action consideration set, increasing policy complexity manifests as movements along the reward-complexity frontier, whereas incomplete action consideration sets manifest as deviations from the frontier. Increasing either would incur opportunity costs in terms of RT. The more important implication is the interplay between the two resource constraints. First, under partial action consideration sets, increasing policy complexity still confers greater trial-averaged reward, but also results in greater suboptimality compared to what could have been achieved under the full action space; this suboptimality can be mitigated by increasing the number of actions considered. Second, the number of actions considered provides an upper bound on policy complexity, which necessitates the increasing of both if the agent hopes to achieve greater reward. The identified relationships provide a nuanced perspective on how action consideration set size and policy complexity may interact synergistically to help agents maintain near-optimal performance.

After identifying the relationships above, we also demonstrate their empirical relevance in explaining human behavior patterns through a contextual multiarmed bandit experiment, varying RT deadline constraints as studied in previous works [20,23]. We found that the framework has its identified relationships reflected strongly in human behavior. Human data supported the framework's predictions on the influence of RT deadlines on policy complexity and the number of actions chosen, the interplay between both constraints that either enlarge or mitigate suboptimality patterns, as well as human efforts to increase both at the same time to reduce such suboptimality. The inclusion of action consideration sets pushes our framework towards a more algorithmic account of human behavior, which is a small step towards addressing criticisms of using vague, abstract concepts such as "information" common to normative frameworks [64], and achieving more authentic characterizations of human decision making. Future works could develop process-level accounts of how humans decide to incorporate more actions (discrete) or to instead refine preexisting state-action mappings (increasing policy complexity alone; continuous), as well as how both quantities are dynamically determined during learning. It would also be valuable to investigate whether action consideration set size and policy complexity compete for a shared pool of cognitive resources, or operate under separate capacity constraints analogous to the slot-continuous dichotomy in working memory [31,65].

Continuing the logic of describing human data, it is also clear that policy complexity and partial action subsets do not fully explain the suboptimality of human participants, because not all of them lie on some $N_a$-specific reward-complexity frontier. This is especially prominent when policy complexity is low and the number of actions chosen is high, forming a cluster of highly suboptimal participants. A potential additional source of suboptimality is the veracity of the agent's $Q(s, a)$ representations. Given the trial-limited nature of our training blocks, an agent who hopes to learn the $Q$-values of more actions must inevitably see each state-action pair less, resulting in noisier learned $Q$-values. This process-level learning problem cannot be captured by the current framework. To assess the influence of noisy $Q$-values on suboptimal behavior, we ran additional simulations that apply Gaussian noise with increasing magnitudes on the true task's $Q(s, a)$ entries. As noise magnitude increases, we indeed see large deviations from optimality at low policy complexity, and in extreme cases, increasing the action consideration set size could enlarge the suboptimality even more (S9 Fig). Given these results, it is plausible that the suboptimal cluster arises as a result of noisy $Q$-values, which is not captured by the normative policy compression framework. Supporting this interpretation, we also find that even among the near-optimal participant cluster, deviations from $N_a$-specific frontiers tend to increase as $N_a$ grows. This pattern suggests potential

difficulties in maintaining accurate representations of additional $Q(s, a)$ entries in working memory [65]. Such representational noise could diminish the theoretical benefits of considering more actions as prescribed by our normative framework, providing another practical reason why agents might prefer smaller action consideration sets. How to behaviorally separate the influence of such reward noise and policy complexity constraints merits future investigation—a deeper understanding of this distinction could help identify more realistic cognitive resource formulations.

From a broader perspective, our framework and the accompanying behavioral evidence have highlighted the complexity of human thinking and meta-reasoning even in very simple tasks, which do not contain exploitable regularities that would motivate hierarchical thinking or abstractions [66,67]. Across our human participants, a significant portion of them did not make use of the full action space. This suggests that they employed some form of meta-reasoning in reducing the original task to a simpler, more manageable task at the cost of lower trial-averaged reward and greater suboptimalities, while simultaneously learning and implementing their low-level policies. Also, participants have determined their policy complexity level and number of actions chosen in a flexible manner, subject to constraints on RT deadlines and their training performance. Failure to navigate these adjustments correlates with greater suboptimality, as conveyed through the identified cluster of suboptimal participants. Importantly, all the above occurs without any explicit instruction on our behalf to change policy complexity or the number of actions chosen, suggesting the spontaneity of such meta-reasoning. It would be informative to assess whether in other typical reinforcement learning tasks, humans exert similar meta-reasoning and spontaneous problem simplification in a way that could not be captured by conventional cognitive models, hence contributing to unexplained sources of variance that is clearly not random noise [68].

## Methods

### Ethical statement

All experiment participants gave informed written consent, and the Harvard University Committee on the Use of Human Subjects approved the experiment (Protocol number IRB15-2048).

### The policy compression framework

Throughout the paper, we restrict our attention to contextual multi-armed bandit tasks for simplicity. In this setup, the environment contains multiple possible states $s \in \mathcal{S}$ with relative frequencies $P(s)$, and actions $a \in \mathcal{A}$. The trial-averaged reward yielded by taking action $a$ in state $s$ is $Q(s, a)$. The agent's goal is to find the policy $\pi(a|s)$—a probabilistic mapping from states to actions—that maximizes trial-averaged reward across all states:

$$V^{\pi} = \sum_{s} P(s) \sum_{a} \pi(a|s) \, Q(s, a). \tag{4}$$

For a resource-rational agent, we formalize their cognitive cost as the mutual information $I^{\pi}(S; A)$ between states $s \in \mathcal{S}$ and the policy-assigned actions $a \in \mathcal{A}$:

$$I^{\pi}(S; A) = \sum_{s} P(s) \sum_{a} \pi(a|s) \log \frac{\pi(a|s)}{P(a)}, \tag{5}$$

where $P(a) = \sum_{s} P(s)\pi(a|s)$ is the marginal probability of choosing action $a$.

We assume that agents are subject to a capacity limit, $C$, which bounds their policy complexity from above. Shannon's noisy channel theorem states that the minimum expected number of bits to transmit a signal across a noisy information channel without error is equal to the mutual information. Therefore, if mapping each state to its best action requires more information that what the agent could afford, the agent must *compress* its policy, or render it less state-dependent. We can therefore define the optimal policy, $\pi^*$, as:

$$\pi^* = \underset{\pi}{\text{argmax}} \, V^\pi, \text{ subject to } I^\pi(S;A) \le C. \tag{6}$$

We can express the above constrained optimization problem in the following Lagrange form:

$$\pi^* = \underset{\pi}{\text{argmax}} \, \beta V^\pi - I^\pi(S;A) + \sum_s \lambda(s) \left( \sum_a \pi(a|s) - 1 \right) \tag{7}$$

where $\beta \ge 0$, $\lambda(s) \ge 0 \; \forall s \in \mathcal{S}$ are Lagrange multipliers, in which the $\lambda(s)$ terms ensure proper policy normalization. Solving this equation yields the following optimal policy:

$$\pi^*(a|s) \propto \exp[\beta Q(s,a) + \log P^*(a)] \tag{8}$$

$$P^*(a) = \sum_s \pi^*(a|s) \, P(s), \tag{9}$$

Where the value of the Lagrangian multiplier $\beta$ depends on the capacity limit $C$. Notably, its inverse is the slope of the reward-complexity frontier at its corresponding policy complexity level $I^\pi(S;A) = C$.

$$\beta^{-1} = \frac{dV^\pi}{dI^\pi(S;A)}. \tag{10}$$

Since the reward-complexity frontier depends on $P(s)$ and $Q(s,a)$, the precise mapping between $\beta$ and $C$ is also task-dependent with no general analytical form [24]. Empirically, the optimal policies at various $C$ can be found by iteratively updating a randomly initialized policy $\pi(a|s)$ using Eq 8 and its marginal $P(a) = \sum_s \pi(a|s) \, P(s)$ until convergence, using a grid of corresponding $\beta$ values. This numerical optimization process is known as the Blahut-Arimoto (BA) algorithm [44,45]. BA-like processes have also been proposed as process-level models of policy compression in humans [24,25].

While the BA algorithm is formally defined for discrete state and action spaces, it can be naturally extended to continuous spaces through appropriate discretization. However, since our study focuses on sampling actions into consideration sets, the resulting action consideration set remains discrete by definition, avoiding the need for discretization. Future work could extend the framework to continuous action spaces through adaptive discretization techniques and investigate corresponding patterns in human behavior.

**Connections to other frameworks.** The optimal policy derived from policy compression shares similarities with the policy prescribed by Kullback-Leibler (KL) regularized control [69], but there are key differences in how each framework handles action distributions and optimization constraints.

In KL-regularized control, a predefined default action distribution—typically uniform over actions—is assumed. Deviations from this distribution are penalized as a cost (negative reward), with $\beta$ in Eq 2 interpreted as a cost sensitivity parameter. In contrast, our framework replaces this fixed distribution with a flexible marginal action distribution, $P^*(a)$, and

formulates the problem as a constrained optimization with $I^\pi(S;A) \leq C$, where $\beta$ acts as a Lagrange multiplier.

A distinguishing feature of policy compression is the dependence of $P^*(a)$ on the policy $\pi^*(a|s)$. This property is central to its normative implications and behavioral predictions. Specifically, it predicts a tendency toward perseveration, where agents favor actions they have frequently chosen in the past—especially when policy complexity is low ($\beta$ is small), making $P^*(a)$ the dominant term in Eq 2. In contrast, KL-regularized control enforces policy regularization toward a fixed default distribution that is independent of the policy, not optimized for individual tasks, and unable to account for perseverative behavior that exploits task regularities.

## Self-normalized importance sampling (SNIS)

Importance sampling is a Monte Carlo method for evaluating properties of a target distribution, using independent samples drawn from a potentially different proposal distribution sharing the same support. Self-normalized importance sampling (SNIS) is a variant of importance sampling that works when the target distribution's density function $p(x)$ is only known up to a normalization constant [49,50].

In SNIS, we assume that independent samples $(x_1, x_2, \dots x_n)$ are drawn with replacement from a proposal distribution $q(x)$. We also know the probability distribution $p(x)$ up to a normalization constant (with its unnormalized density function denoted as $\rho(x)$). Our goal is to estimate the mean $\mathbb{E}_p[f(X)]$ of some transformation $f(X)$ under $X \sim p(x)$. To do so, we construct importance weights $w_i = \rho(x_i)/q(x_i)$ for each sample $x_i$, and use the estimator $\hat{\mu}$:

$$\mathbb{E}_p[f(X)] \approx \hat{\mu} = \frac{\sum_{j=1}^{n} w_j f(x_j)}{\sum_{j=1}^{n} w_j} = \frac{\sum_{j=1}^{n} \frac{\rho(x_j)}{q(x_j)} f(x_j)}{\sum_{j=1}^{n} \frac{\rho(x_j)}{q(x_j)}}. \tag{11}$$

The estimator $\hat{\mu}$ is biased under finite sample size $n$, due to the quotient operation involving random variables $w_j$. However, it is asymptotically unbiased in the limit of infinite sample size [49,50].

To see how SNIS applies to policy compression, we rewrite Eq 8 using the unnormalized optimal policy $\eta^*$ for some $\beta$ value (corresponding to some capacity limit $C$):

$$\eta^*(a|s) = \exp\left(\beta Q(s,a)\right) P^*(a). \tag{12}$$

Next, we draw independent action samples $(\alpha_1, \dots \alpha_n) \in \mathcal{A}^n$ from the proposal distribution $P_0(a)$. We can estimate the unnormalized policy's assigned unnormalized probability value for every $(s,a)$ pair:

$$\eta^*(a|s) \approx \sum_{j=1}^{n} \exp\left(\beta Q(s,\alpha_j)\right) \frac{P^*(\alpha_j)}{P_0(\alpha_j)} \mathbb{I}(\alpha_j = a), \tag{13}$$

where $\mathbb{I}(\alpha_j = a)$ is the indicator function comparing $\alpha_j$ to $a$. Hence, the optimal policy's assigned probability to this $(s,a)$ pair can be estimated as:

 

$$\pi^*(a|s) := \frac{\eta^*(a|s)}{\sum_{a_i \in \mathcal{A}} \eta^*(a_i|s)} \tag{14a}$$

$$\approx \frac{\sum_{j=1}^n \exp\left(\beta Q(s, \alpha_j)\right) \frac{P^*(\alpha_j)}{P_0(\alpha_j)} \mathbb{I}(\alpha_j = a)}{\sum_{a_i \in \mathcal{A}} \sum_{j=1}^n \exp\left(\beta Q(s, \alpha_j)\right) \frac{P^*(\alpha_j)}{P_0(\alpha_j)} \mathbb{I}(\alpha_j = a_i)} \tag{14b}$$

$$= \frac{\sum_{j=1}^n \exp\left(\beta Q(s, \alpha_j)\right) \frac{P^*(\alpha_j)}{P_0(\alpha_j)} \mathbb{I}(\alpha_j = a)}{\sum_{j=1}^n \exp\left(\beta Q(s, \alpha_j)\right) \frac{P^*(\alpha_j)}{P_0(\alpha_j)}} =: \hat{\pi}^*(a|s) \tag{14c}$$

with Eq 14c stemming from the fact that each $\alpha_j$ must take on some value in $\mathcal{A}$.

Notice that Eq 14c takes the form of the SNIS expression in Eq 11—with $f(X) := \mathbb{I}(\alpha_j = a)$, $q(x) := P_0(a)$, and $\rho(x) := \exp\left(\beta Q(s, \alpha_j)\right) P^*(\alpha_j)$. Hence, $\hat{\pi}^*(a|s)$ is an asymptotically unbiased estimator of the optimal policy $\pi^*(a|s)$ at the same $\beta$ value under the full action space. We can hence use $\hat{\pi}^*(a|s)$ for action selection.

The Blahut-Arimoto algorithm can be used to obtain the estimator $\hat{\pi}^*$ by iteratively applying Eq 14 and $P(a) = \sum_s P(s)\,\pi(a|s)$ on a randomly initialized policy $\pi$ until convergence. Now across different sample sizes $n$, we can assess the average deviation of $\hat{\pi}^*$ (computed for each action sample $(\alpha_1, ..., \alpha_n)$) from the true task's reward-complexity frontier.

## Simulation details

We ran 200 random simulations for each Langrange multiplier $\beta$ value within a grid, where randomness is introduced by the underlying action proposal distribution $P_0(a)$. The process is repeated for each action proposal distribution, action consideration set size (for sampling without replacement; $N_a$) or action sample size (for sampling with replacement; $n$), as well as the three valid combinations of sampling methods (with or without replacement) and bias-correcting algorithms. In Figs 3 and S1–S5, we visualize 2D errorbars showing mean±SEM of policy complexity and reduction in trial-averaged reward, aggregated over the 200 simulations for the same $\beta$.

## Human experiment details

**Participants.** One-hundred-and-one participants (75 women, 25 men, 1 prefer not to say) were recruited. We selected the sample size based on the reaching of statistical significance in all planned analyses in a separate group of $N = 30$ pilot participants (data excluded from final analysis). All analyses were preregistered at https://aspredicted.org/s44z-xf4p.pdf unless otherwise noted. We excluded 26 participants for not responding within the response time (RT) deadline for more than or equal to 20 trials across all test blocks, leaving data from 75 participants (52 women, 23 men) for subsequent analyses. Participants were paid a base pay of $6 and a performance bonus of up to $2 for completing the task. Participants took, on average, 27 minutes to complete the entire experiment, and their average payout was $7.43.

**Procedures.** Each participant completed three test blocks containing 96 trials each. On each trial, participants see an image and must press a key (action) before the RT deadline. There were six possible images (states) and seven available actions, which are shared across blocks. Each state was assigned a unique optimal action (one of the six number keys; the mapping is randomized across participants), while the remaining action (the letter key "E") was a "safety" action that guaranteed a smaller reward across all stimuli (Fig 4A–4B). The three blocks featured RT deadlines of 0.5s, 1s, and 2s in randomized order, which are revealed to participants before starting each block. Participants were informed that the mapping from

state to action was held fixed across all blocks, and that they would receive a bonus proportional to their summed performance across blocks.

Reward delivery was deterministic. The safety action that always yields +0.2 reward regardless of the state. The remaining six "unsafe" actions are uniquely optimal for each of the six states, yielding +1 or −0.18 reward according to the state identity. Failure to respond within the RT deadline resulted in a −1 reward penalty and automatic progression to the next trial. After making a response (or when the RT deadline arrived), participants were given immediate feedback for 0.5 s—a border around the image whose color matched the reward value (dark green +1, light green +0.2, pink −0.18, red −1). Then, the numeric reward value was displayed for another 0.5 s, before the next trial's stimulus appeared. Participants could track the total reward earned during the block, displayed as a number above the image. At the end of each block, they were provided with feedback on the total reward they earned in that block (Fig 4C).

Before the three test blocks mentioned above, participants completed four training blocks, with each block lasting 60 trials. These training blocks are not analyzed. The first shared training block had RT deadline 2 s. In this shared training block, participants were provided feedback regarding the optimal action for the preceding stimulus, after their response and before the next trial began. The next three condition-specific training blocks had RT deadlines 2 s, 1 s, and 0.5 s, and no longer contained feedback on optimal actions just like the test blocks. These condition-specific training blocks help participants prepare themselves for the test blocks and develop their key-pressing strategy for each RT deadline. The optimal state-to-action mappings were revealed to participants before the shared training block, the first condition-specific training block, and the first test block.

**Reward-complexity frontiers for different numbers of actions chosen.** The simplicity of the task structure allows us to exhaustively enumerate reward-complexity frontiers of different action consideration sets $N_a$, depending on their size and whether the safety action is included (Fig 4D–4E). The frontiers may end before the maximum policy complexity allowed by the action consideration set size at $\log_2\left(\max(|\mathcal{S}| = 6, N_a)\right)$ bits, as under the frontier's corresponding action consideration set, further increasing policy complexity towards the maximum bound would only confer less trial-averaged reward. We consider the higher of the two reward-complexity frontiers associated with either including or excluding the safety action at each policy complexity level, and visualize it as the $N_a$-specific reward-complexity frontier (Fig 4F). The downward kink for $N_a = 4, 5$ reflects the fact that the reward-complexity frontier associated with including the safety action extends farther towards higher policy complexity, but yields lower trial-averaged reward compared to excluding the safety action. In other words, the frontier after the downward kink reflects a suboptimal action consideration set choice at the same $N_a$ level, but it still has a normative basis in being optimal under that suboptimal consideration set choice.

**Estimating policy complexity and number of actions considered.** We defined policy complexity as the mutual information between the observed states and chosen actions. Following prior work [21,24,25], we estimated the policy complexity of each participant in each ITI condition using the Hutter estimator [70]. Specifically for each state, we assume a symmetric Dirichlet prior with $\alpha = 0.01$ for all actions chosen, and use the empirical action counts to reach a posterior Dirichlet distribution over action probabilities. We then estimate policy complexity as the mutual information of the posterior mean policy. The above procedure is informed by previous literature, reporting that the resulting estimates exhibit reasonably good performance when the joint distribution is sparse [71]. The choice of $\alpha = 0.01$ is informed by

rate-distortion theory, stating that empirical trial-averaged reward values cannot be above the reward-complexity frontier. We have chosen $\alpha = 0.01$ empirically to obey this property.

To estimate the number of actions considered in a test block by a participant, we use the number of distinct actions chosen by that participant in that block ($N_a$). The estimator was chosen to best compare participant performance to the reward-complexity frontiers of corresponding action consideration set size—if a participant has chosen $N_a = 4$ distinct actions within a block, their performance is upper-bounded by the reward-complexity frontier corresponding to $N_a = 4$. However, to accommodate the possibility that participants brushed over unintended keys, we re-performed all analyses based on stricter thresholds, counting a chosen action into the action consideration set size $N_a$ if and only if the participant has chosen the action for at least two or three times within the block (see Tables B and C in S1 Appendix). None of the conclusions change as a result of thresholding, which highlights the robustness of our findings.

**Statistical analysis.** We fit linear mixed-effects (LME) models to study the relationship among behavioral variables of interest: policy complexity, the number of actions chosen, RT, trial-averaged reward, and loss in trial-averaged reward. We obtained parameter estimates using maximum likelihood estimation with the "fitlme" function in MATLAB R2023a. In the main text, we report the fitted coefficients and $p$-values for fixed effects of interest, as well as the standard deviation of their corresponding random effects.

We also computed Pearson correlation coefficients between pairs of behavioral variables: policy complexity, number of actions chosen, and training block accuracy rate. In the main text, we report the coefficients and their $p$-values.

Based on all our predictions, we have performed the following analyses:

```
1) PolicyComplexity ~ RTDeadlineCond + (RTDeadlineCond|Participant);
2*) Reward ~ RTDeadlineCond + (RTDeadlineCond|Participant);
3) P(a_safety) ~ RTDeadlineCond + (RTDeadlineCond|Participant);
4) RT ~ PolicyComplexity + (PolicyComplexity|Participant);
5) Na ~ RTDeadlineCond + (RTDeadlineCond|Participant);
6) RT ~ Na + (Na|Participant);
7) Reward ~ PolicyComplexity*Na + (PolicyComplexity*Na|Participant);
8) RewardLoss ~ PolicyComplexity*Na + (PolicyComplexity*Na|Participant);
9) Correlation: PolicyComplexity and Na;
10) Correlation: TrainAccuracy and PolicyComplexity;
11) Correlation: TrainAccuracy and Na;
12*) RT ~ PolicyComplexity*Na +(PolicyComplexity*Na|Participant);
```

where the asterik ($*$) denotes post-hoc analyses. The above numeric indexing of statistical analyses is also used in S1 Appendix.

## Supporting information

**S1 Appendix. Tables of human behavioral results under different action counting thresholds.**
(PDF)

**S1 Fig. Simulating the trial-averaged reward of partial action consideration sets, in a random reward structure task.** Rows structure is identical to Fig 3. Columns denote trial-averaged reward instead of loss in trial-averaged reward.
(TIFF)

**S2 Fig. Simulating the trial-averaged reward of partial action consideration sets, in a task where reward is centered on few actions.** Row structure is identical to S1 Fig.
(TIFF)

**S3 Fig. Simulating the suboptimality of partial action consideration sets, in a task where reward is centered on few actions.** Row and column structure are identical to Fig 3.
(TIFF)

**S4 Fig. Simulating the trial-averaged reward of partial action consideration sets, in the human experiment task.** Rows and column structures are identical to S1 Fig.
(TIFF)

**S5 Fig. Simulating the suboptimality of partial action consideration sets, in the human experiment task.** Rows and column structures are identical to Fig 3.
(TIFF)

**S6 Fig. Relationships between pairs of human behavioral variables.** Relationships between **(A)** number of actions chosen and policy complexity (black solid line denotes maximum policy complexity enabled by each number of actions chosen), **(B)** policy complexity and RT, **(C)** number of actions chosen and RT, **(D)** policy complexity and probability of choosing the safety action. Color denotes the RT deadline condition, and semitransparent gray lines connect the same participant's data. Training and test block relationships include **(E)** training block mean accuracy and test block mean policy complexity, and **(F)** training block mean accuracy and test block mean number of actions chosen. Color denotes the participants we have included or excluded based on the reward > 0.15 cutoff.
(TIFF)

**S7 Fig. Policy complexity distributions for each number of actions chosen.** Each panel corresponds to the distribution of empirical policy complexity for a particular number of actions chosen ($N_a$), across all corresponding participants and RT deadline conditions. The $x$–axis is scaled to reflect the allowable range of policy complexity( between0 and $\log_2(N_a)$ bits). $N_a = 1$ was excluded because it only allowed a fixed policy complexity of 0 bits.
(TIFF)

**S8 Fig. Action distributions for each participant. Row 1:** Each participant's empirical action distribution $P(a)$ over all action keys (semitransparent lines; color denotes RT deadline condition), as well as their mean $\pm$ SEM (black errorbars). **Row 2:** Same as Row 1, but the actions are reordered for each participant based on their relative frequency. The black horizontal dotted line denotes chance probability of 1/7.
(TIFF)

**S9 Fig. Simulating the influence of noisy $Q$-values.** The human experiment task's $Q(s, a)$ values are smeared with Gaussian noise with increasing standard deviation (stratified into columns). **Row 1:** Assuming that the safety action is retained in the consideration set, the influence of policy complexity (x-axis) and action consideration set size (color) on trial-averaged reward (y-axis). The full-action-space reward-complexity frontier is depicted as a black line. 2D errorbars denote mean±SEM of policy complexity and trial-averaged reward over 200 random simulations, aggregating over simulations that share the same $\beta$. **Row 2:** Same as Row 1, but Assuming that the safety action is not retained in the consideration set.
(TIFF)

## Acknowledgments

We thank Arthur Prat-Carrabin, Bilal A. Bari, and Rahul Bhui for helpful discussions and feedback.

## Author contributions

**Conceptualization:** Shuze Liu, Samuel Joseph Gershman.

**Data curation:** Shuze Liu.

**Formal analysis:** Shuze Liu.

**Funding acquisition:** Samuel Joseph Gershman.

**Investigation:** Shuze Liu.

**Methodology:** Shuze Liu, Samuel Joseph Gershman.

**Project administration:** Shuze Liu, Samuel Joseph Gershman.

**Resources:** Samuel Joseph Gershman.

**Software:** Shuze Liu.

**Supervision:** Samuel Joseph Gershman.

**Visualization:** Shuze Liu.

**Writing – original draft:** Shuze Liu.

**Writing – review & editing:** Shuze Liu, Samuel Joseph Gershman.

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
