## [Decision Letter · Decision Letter 0]

11 Mar 2025

PCOMPBIOL-D-24-02072

Action subsampling supports policy compression in large action spaces

PLOS Computational Biology

Dear Dr. Liu,

Thank you for submitting your manuscript to PLOS Computational Biology. We have received two reviews from experts in the field. I am in full agreement with both reviewers that the paper is interesting, well motivated, and ambitious. I also, like reviewer 2, think that this paper would have a wide appeal within the cognitive science community. I think the both reviewers raise reasonable points, and I won't repeat them all here, but I think addressing these points will make for a much stronger paper. 

I will, however, emphasise the points of reviewer 2 that the writing in this paper could be made much clearer. I would advise thinking about reducing jargon, reducing repeated information across sections, and using consistent wording. Reviewer 2 gives several such examples, and I'll provide a few of my own: On page 2 of the ms., there is a reference to Fig.1A. However, the figure includes several terms that are defined later in the text. When I first read the paper I spend a couple of minutes trying to figure out what I had missed. I think it would be much clearer to just say, "These findings have inspired a descriptive... state-dependent action evaluation, which we present below". 

The term "general-value-based action is used on line 29 before it is defined. On lines 201, and 211 the terms "general-value-guided action sampling" and "general-value-based sampling" are used. Are these meant to be different concepts? If not, why use different terms? 

On line 92, an equation is presented, and the term Q(s,a) is not defined until line 117, after another equation using the term is presented, and in a completely different paragraph. 

Again, I emphasise that it is only because I think this paper would be of interest to a wider audience that I pick these nits. I hope that these and the reviewer comments help you to revise your ms. 

Please submit your revised manuscript within 60 days May 11 2025 11:59PM. If you will need more time than this to complete your revisions, please reply to this message or contact the journal office at ploscompbiol@plos.org. Please include the following items when submitting your revised manuscript:

We look forward to receiving your revised manuscript.

Kind regards,

Alex Leonidas Doumas

Academic Editor

PLOS Computational Biology

Andrea E. Martin

Section Editor

PLOS Computational Biology

**Journal Requirements:**

2) Thank you for including an Ethics Statement for your study. Please include:

i) A statement that formal consent was obtained (must state whether verbal/written). NOTE: If child participants, the statement must declare that formal consent was obtained from the parent/guardian.].

Potential Copyright Issues:

i) Figures 4A, and 4C. Please confirm whether you drew the images / clip-art within the figure panels by hand. If you did not draw the images, please provide (a) a link to the source of the images or icons and their license / terms of use; or (b) written permission from the copyright holder to publish the images or icons under our CC BY 4.0 license. Alternatively, you may replace the images with open source alternatives. See these open source resources you may use to replace images / clip-art:

**Reviewers' comments:**

Reviewer's Responses to Questions

Reviewer #1: Action subsampling supports policy compression in large action spaces

Recommendation: Minor revisions.

Overall the paper is well structured in terms of introducing the main ideas of the paper, presenting simulations that motivate these ideas, and concluding the human subject experiments to make concrete determinations related to the questions under consideration. However in terms of the overall motive there are some things that are less clear, this is the majority of the comments on specific lines that I have raised below. I think that the background and theoretical motivation could benefit from a discussion of the similarities and differences between this proposed method and information-constraints in continuous action spaces in RL. Another connection that I would like to be clarified is between selecting action consideration sets and MARL settings where one agent’s action follows another agent, this is especially relevant in settings where there is an information asymmetry between agents. If these two areas of research are not relevant then a short description of them and why they are not related would benefit the manuscript. Many of the points below are clarifications and if they can be resolved in an author reply then they need not necessarily be added to the manuscript itself.

Lines 49-50: An agent that is able to deliberately meta-reason about how best to construct consideration sets seems to necessarily have either a higher capacity or more time to consider how best to construct the consideration set before selecting an action. This is highly distinct from the case where an agent is compelled by resource limitations to simplify the rate-distortion problem over a reduced consideration subset. It is unclear from the introduction whether the proposed method is intended to handle one of these specific cases or both.

Lines 115-116: I don’t believe that the traditional formulation of resource rational policy compression assumes that the agent has perfect access to the ground truth P(s) and Q(s,a). In my experience methods that have utilized policy compression in understanding capacity-constrained decision making in humans, the source of the P(s) and Q(s,a) terms comes from the experiential state probability and state-action values of an individual. These values are not supplied omnisciently from the task but estimated based on agent experience.

Line 133: Is this meant to suggest that P(s) and Q(s,a) are generally symmetric or only symmetric under the equal state probability and uniquely optimal state-action values.

Fig 2: Is there a difference between the theoretical ‘frontier corresponding to the full action space’ in black and the Na = 6 frontier? I don’t see why both are included.

Line 163-164: I am having a hard time conceptualizing how or why an action would be sampled multiple times in the construction of a consideration set, as adding it to the set multiple times versus a single time seems meaningless to me. Is the importance of distinguishing between with and without replacement related to the necessity of keeping the current consideration set values in memory while sampling? I understand how repeated sampling can impact the estimate of outcomes, but in constructing consideration sets I am less clear.

Lines 271-291: The second conclusion of these simulations appears obvious, perhaps a more thorough connection to either hierarchical or continuous action space RL as suggested above can clarify why this is not the case. By this point in the manuscript I was under the impression that the main interest was in how humans handle the balance of complexity in the construction of consideration sets and the optimization of behavior under the dual constraints of information processing and limited action consideration. The final lines of this section in 288-291 further confuse the overall goal of this investigation.

Line 302: I performed the online experiment and perhaps it was because I had read most of the paper already but I didn’t find myself actively forming a consideration set. Has this environment been used in studies of human decision making in action considering set formation? Is the idea that if I was only able to remember the first three stimuli that I would only consider the first three actions and then default to the safety action for the other three? Because in that case there seems to be an effect on the state-action relationship as the agent is effectively reducing the three remaining states to a single one with a higher probability of occurring. I am not sure if I can see the direct connection between this idea of necessitating the construction of consideration sets through the use of a ‘safety’ action and the exact structure of the simulations as they did not have this safety action.

Lines 331-333: Predictions 5 and 6 seem like they could both be caused by either smaller consideration sets alone, less complex policies alone, or a combination of these two factors. The way the experiment is constructed makes it seem hard to disentangle these two effects which I had assumed was the main interest of the manuscript. That time constraints lead to either smaller consideration sets OR less complex policies seems obvious and necessary from the way the task is structured, but what I am most interested in is how people optimize the balance between these, particularly whether they are working with a single constraint on both complexities or separate processes with independent constraints.

Lines 445-449: One difficulty is the binary distinction between generally-valuable action sampling and uniform sampling. I can see that choosing between these two would be based on the complexity of the policy, but it doesn’t seem like it needs to be strictly one or the other. I would imagine that there would be a way to optimize the complexity of the sampling distribution relative to a specific capacity on behavior complexity.

Reviewer #2: Liu & Gershman develop a theoretical framework for decision-making in situations where there are a large set of actions and consequently a large set of state-action mappings. Since humans have limited cognitive resources, they struggle to retain all state-action mappings when actions become large. As a consequence, humans make sub-optimal decisions. In situations where decisions are related to rewards, humans fail to earn the maximum amount of (average) reward in the task. The goal of this paper is to characterise this sub-optimality. In particular, the authors are interested in breaking down the sub-optimality along two (orthogonal) dimensions: "policy complexity" and number of actions in an "action consideration set size". Policy complexity is defined in information-theoretic terms but can be intuitively understood as the degree to which (learned or behaviourally exhibited) mappings between states and actions is deterministic. Action consideration set size is the number of actions that an agent chooses from. As the number of possible actions increases, this set size is characterised by a compromise made by the decision-maker, discarding some actions from the set due to limitations in cognitive capacity.

There are many things to like about the paper. The theoretical development is rigorous, it is connected with a set of predictions which are then tested in a set of experiments. The goal of the paper is ambitious but important - how to characterise human decision-making in real-world scenarios where the state-action mappings are complex and hard to learn due to the large number of possible actions.

However, ultimately I was left quite frustrated by the paper. I am not familiar with the theoretical framework that the authors are trying to extend (though I think I understand the gist of it), so I am going to focus here on the empirical contribution of the paper and it's writing. Let me address the writing first. I feel authors can do a lot more with the way the paper is written. In it's current form, the paper is going to be comprehensible to very few people in the field. The writing style is cryptic, full of jargon and the authors have made very little effort to make it comprehensible to the average psychologist.

Here is an example of a typical phrase from the paper:

"Assuming running BA on the remaining action consideration set, our predictions are supported: the general-value-based proposal distribution conferred a clear advantage at low policy complexity and small action consideration set size, compared to the uniform proposal distribution. This manifested as proximity to both the oracle proposal distribution as well as the x=0 line, which indicates closer overlap with the full-action-space reward complexity frontier and hence less suboptimality (Fig 3B Row 1)."

What the authors presumably mean is that the reduction in trial-averaged reward was lower for a simulated agent that sampled actions from a general-value-based distribution compared to an agent that sampled actions from a flat distribution. The reduction in trial-averaged reward was smallest for an agent that sampled actions from an oracle distribution. Instead the authors seem to rely on a much more cryptic language and I still don't understand why the authors talk about an "overlap" with the reward-complexity frontier.

Here's another example from the section on Bias-correction algorithms:

"Notably, SNIS does not guarantee optimal performance under the current action consideration set at all policy complexity levels (whereas BA does); instead, it focuses on representing the full-action-space optimal policy at the same β without any bias, a goal whose empirical utility is subject to assessment."

Again, this strikes me as really cryptic language. Could the authors clarify why SNIS does not guarantee optimal performance under the current action consideration set. Also why is the empirical utility subject to assessment. More importantly, the authors haven't made it clear what motivates them to explore "bias-correcting" algorithms.

There are many other examples throughout the manuscript. It seems to me that the target audience for the authors is a very limited set of researchers and a lot of it will not make sense to the general Cognitive Psychologist.

Let me now turn to the empirical aspects of the paper. It is again difficult for me to understand what specific empirical phenomenon the author wanted to investigate. It occurred to me that estimating the policy complexity that a participant was using would be a very useful insight into their cognitive processes. However, it turns out that policy complexity is estimated based on previous research. This is what the authors write about estimating policy complexity (pg 20):

"Following prior work [21,24,25], we estimated the policy complexity of each participant in each ITI condition using the Hutter estimator, which computes the posterior mean value of mutual information under a symmetric Dirichlet prior [62]"

Again, it would have been really useful if the authors gave some insight into how this policy complexity is estimated by this algorithm. I turned to reference [20] to gain some insight into this. This is what is stated there:

"estimation of mutual information is notoriously tricky (see Paninski, 2003). We used the Hutter estimator, which computes the posterior expected value of the mutual information under a Dirichlet prior (Hutter, 2002). We chose a symmetric Dirichlet prior with a concentration parameter *α* = 0.1, which exhibits reasonably good performance when the joint distribution is sparse (Archer, Park, & Pillow, 2013)."

So, not much more information there. I turned to reference [24]. This is what is stated there:

"Empirical policy complexity ... was estimated from each subject’s behavior per task condition. Following [12], we use the Hutter estimator, which computes the posterior expected value of the mutual information under a symmetric Dirichlet prior [44], to estimate subjects’ empirical policy complexity, or the mutual information between the observed stimuli (states) and subjects’ key press responses (actions)."

In other words, all the references use the same language and the reader can't understand what exactly was done to estimate the complexity. While I can infer that the authors estimated the policy complexity as the mean of a posterior (multinomial?) distribution using the Dirichlet prior, it would be really useful to have a bit more information about it. For example, in Figure 5 (the key Results figure) the policy complexity for each participant is shown as a dot. I would have liked to know what is the confidence in each point estimate. Without any further details and despite checking cited material I'm still not sure about this.

The other contribution of the paper could have been to understand participants' "action consideration sets." However, this again turns out to be not one of the key contributions as the authors simply use a rough estimate for this (pg 21):

"To estimate the number of actions considered in a test block by a participant, we use the number of distinct actions chosen by that participant in that block (Na)."

How does one know that participants used the same consideration set for each decision in a block?

Perhaps the real contribution of the paper is to work out the theoretical frontier in the reward-complexity space. That would be fine if one was only interested in the normative model. But it seems to me that the authors would like to make an empirical point. But when one compares participants with the theoretical frontier, there are many differences. Not only is there a cluster of participants who seem nowhere near the frontier (as the authors discuss), even the participants who are close to the frontier systematically deviate from the frontiers for middle values of policy complexity (between 0.5 and 2). Again, it would have been useful here to know what the confidence of estimated policy complexity is in each case.

Of course, the authors make a number of empirical predictions (11 in all). But it turns out that most of these are not predictions from the theoretical work in this paper but from previous papers (predictions 1-6). In fact, there is nothing in the theory presented in the current paper that makes a link to RTs. And it is hard for me to see how any of these predictions are strong tests for falsifying models. I will not enumerate all of them, but in simple terms, this is what some of the predictions say:

- when response deadlines are short, participants' mapping between states and rewards will be more random (prediction 1)

- when response deadlines are short, participants will accumulate less reward (prediction 2)

- when response deadlines are short, participants are more likely to choose the safety action (prediction 3)

- when participants use a more deterministic mapping between states and actions (i.e. when participants have a better idea of which state maps to which action), they end up earning larger reward (prediction 7) and this reward becomes even higher for participants that can map more states to actions compared to participants that can map less states to action

- participants who show higher accuracy in training blocks end up being participants who are, on average, better at mapping states to actions (prediction 10).

Authors, please correct me if I'm wrong, but it seems to me that any reasonable model should make all of these predictions - so they don't seem to be predictions that will help us adjudicate between different models.

To sum up, I think that the paper has a lot of promise and the research direction is both interesting and innovative, but in its current form the paper will be incomprehensible to many cognitive psychologists and it also fails to distinguish what it's key (empirical) contribution is. I suggest the authors rewrite it to improve clarity, perhaps even moving some of the technical details to an Appendix, and rethink how they want to state their contributions.

Minor points:

lines 167-167: After sampling actions from the proposal distribution P0(a). The simplest solution -> there should be no full stop

line 177: "and hence feel that" -> and hence *we feel that?

line 324: arise -> arises

Figure 5I: Is it possible for all complexity values to have all number of actions. If not, would a Pearson correlation be the correct measure of correlation?

line 491: "action consideration set considerations pushes" -> extra considerations

line 512: "As noise magnitude increases, we indeed see large deviations from optimality...": So can participants lie anywhere in the reward-complexity space and still be compatible with the model here? What kind of participant behaviour will falsify the model then?

**Have the authors made all data and (if applicable) computational code underlying the findings in their manuscript fully available?**

Reviewer #1: Yes

Reviewer #2: Yes

PLOS authors have the option to publish the peer review history of their article (what does this mean?). If published, this will include your full peer review and any attached files.

Reviewer #1: No

Reviewer #2: No

**Figure resubmission:**
---

## [Decision Letter · Decision Letter 1]

26 May 2025

PCOMPBIOL-D-24-02072R1

Action subsampling supports policy compression in large action spaces

PLOS Computational Biology

Dear Dr. Liu,

Thank you for submitting your manuscript to PLOS Computational Biology. Thank you for submitting your revised manuscript. I agree with the reviewers that the revised version is much clearer than the previous version. You will see that reviewer 2 has raised a few additional points to help the clarity of the manuscript. I think addressing these points will indeed benefit the paper. Though, of course, I cannot make any guarantees, I do not foresee having to send this paper out for review again once revisions are made.

Please submit your revised manuscript within 30 days Jul 26 2025 11:59PM. If you will need more time than this to complete your revisions, please reply to this message or contact the journal office at ploscompbiol@plos.org. Please include the following items when submitting your revised manuscript:

We look forward to receiving your revised manuscript.

Kind regards,

Alex Leonidas Doumas

Academic Editor

PLOS Computational Biology

Andrea E. Martin

Section Editor

PLOS Computational Biology

**Journal Requirements:**

1) We note that your Supplementary Figures files are duplicated on your submission. They are uploaded as separate figure files and included in the supporting information file. Please remove any unnecessary files from your revision, and make sure that only those relevant to the current version of the manuscript are included. 

2) Please ensure that all Figure files have corresponding citations and legends within the manuscript. Currently, Figure 4 in your submission file inventory does not have an in-text citation. Please include the in-text citation of the figure.

3) Please amend your detailed Financial Disclosure statement. This is published with the article. It must therefore be completed in full sentences and contain the exact wording you wish to be published. Since the funders had no role in your study, please state: "The funders had no role in study design, data collection and analysis, decision to publish, or preparation of the manuscript."

**Reviewers' comments:**

Reviewer's Responses to Questions

Reviewer #1: Regarding my own comments, I am now much clearer on the background, motivation, and claims made in the paper thanks to the authors added clarity in the manuscript and additional connections to previous literature. The authors went line by line for all of the comments I had made in the manuscript, even minor clarification ones, which has fully addressed all my concerns. I think the current version of the paper is much stronger and I don't have any other comments or questions regarding the content. I only have one quibble below but it is fairly minor.

On line 103 perseveration is defined as 'state-independent actions' but to me that term is slightly different, but this isn't a big issue. But that line also references Figure 1.C which I don't think shows a state-independent policy since that's just the equation. Figure 1.D almost shows a state-independent policy but even there the probability of selecting action a1 is higher. Maybe making that distribution flat and saying (left side Figure 1.D) would be a better example of state-independent actions, or some other way.

Since I was perhaps overly familiar with the background of the paper I had not considered whether it would be easily readable by the average psychologist. These comments were raised by Reviewer 2 and I have looked through the concerns and how they are addressed. At this point it is still difficult for me to evaluate how well this has been achieved but I do agree with the author's attempts to make the paper more readable for a general audience. I also find it difficult to notice typographic mistakes like the ones raised by reviewer 2 so I will leave it to them and the editor to confirm the improvements to grammar and spelling.

Reviewer #2: I appreciate the changes made by the authors to improve clarity and accessibility of the manuscript. I think these changes have been positive and have made me appreciate their work a bit more. While the manuscript is definitely in a better shape, there are a few remaining concerns that, in my view, should be addressed:

1. The sections that I still found difficult to parse is the one on 'Bias-correction algorithm' (pg 7). The goal here seems to be comparing the Blahut-Arimoto algorithm with an algorithm that does not produce a biased estimate. The authors haven't spent much time in the manuscript motivating this goal, i.e. discussing how much of a problem generating a biased estimate could be. It strikes me that the manuscript could benefit from simulations showing how a bias in P_0(a) affects the results of the Blahut-Arimoto algorithm before considering alternative models (if alternative models are to be presented)

Moreover, the alternative model (SNIS), which produces an unbiased estimate, seems like a strawman (or, at least, an afterthought). Unsurprisingly, this model is rejected a few pages later (pg 9-10). Furthermore, SNIS only produces an unbiased estimate "in the asymptotic limit of infinite samples". So, it's not really a cognitively plausible model. If SNIS is not critical to the author's thesis, perhaps a better place for it will be the Appendix. That will further increase the accessibility of the manuscript.

2. Related to point 1, the manuscript doesn't spend much time discussing the results of SNIS algorithm in Figure 3B. These results, at least at first glance, seem strange. While the BA algorithm (top row) predicts that an increase in policy complexity leads to better trial-average reward -- which makes sense -- SNIS predicts a non-monotonic relationship between policy complexity and reward. That is, sometimes increasing complexity leads to better trial average reward, while other times it leads to worse trial average reward. Similarly, the comparison between proposal distributions leads to to some odd results here. In the case of the BA algorithm sampling from the oracle distribution performs better than the value-based action sampling, which performs better than uniform action sampling. This makes sense. But for SNIS, there is no clear relationship. For some policy complexity values the value-based sampling performs better than uniform action sampling, while for other values the relationship reverses. Moreover, for some policy complexity values, both value-based and uniform sampling perform /better/ than the oracle distribution. I don't understand why this is the case and I can't see an explanation in the manuscript for these effects. If the authors think all this does not need to be discussed, then this again speaks for moving SNIS to the appendix.

3. Pearson correlation: I'm afraid I don't buy the author's defence of using Pearson's r to measure the correlation between number of actions chosen and policy complexity, in Figure 5I (see my comments and author's response). I understand that the authors are arguing that it is theoretically possible to have zero correlation and therefore having a correlation of 0.67 is significant. But is this a strong or a weak correlation? The authors suggest that this correlation is "positive and strong" (line 442). But I'm not so sure.

Here's how I'm thinking about it: When number of actions are 2, the policy complexity can only vary from 0 to log(2). When number of actions are 6, the policy complexity can vary between 0 and log(6). That is, the range increases as number of actions increase. Because of this, if an agent chose the policy complexity completely randomly from a uniform distribution you will get a correlation between number of actions and policy complexity. So, is a correlation of 0.67 "positive and strong"n? Well, firstly correlation in this case cannot be negative. Also, the correct baseline for the correlation here is not 0. Rather, it is the correlation for this agent who chooses complexities randomly from a uniform distribution. Can the authors compute this before claiming that the correlation is strong?

4. Figure 2B: Could the authors make it clear in the caption (or where they discuss this figure) that policy complexity can only vary between 0 and log(Na). That is why the curves in this figure have different ranges. I remember this was one of my key confusions when I first saw this figure.

5. Figure 2B: The frontier for the full action space is the same as the frontier for Na=6 because there are only six actions in the full action space. Therefore, the green lines and black lines overlap, by definition. So, why plot both these lines on the same graph. Wouldn't it be better to remove the black line and change the legend "Na=6" to "Na=6 (full)" or something similar?

6. lines 155--157: "Building on this motivation... task setups". This is really vague. Building on what motivation? I don't think the authors have motivated exploring the advantages of biased action sampling (see point 1).

7. line 161: "Following the two-step framework", I think this was called the “two-stage model of open-ended decision-making” above. Worth maintaining the same language so as not to confuse the reader. Also, I would suggest that the authors insert a reference to Figure 1A again here to make it clear that they are talking about a concept previously developed in the manuscript.

8. line 174: "orthogonal dimension": Did the authors mean "another model-parameter worth exploring"? This is also the first time the authors mention comparing the biased vs unbiased estimators. But it hasn't been motivated (see point 2 above).

9. line 179: "sample.The" missing space

10. line 181: "Notably BA is agnostic towards the number of times a particular action is sampled". What does it mean for an algorithm to be agnostic? Do you mean that, in principle, BA can be set up in a way that it samples actions with replacement? Again, there is a lot in the manuscript that can be improved with the discussion of bias in BA algorithm.

11. line 260: "x=0" Did the authors mean y=0 line?

12. line 296: "the suboptimality in trial-averaged reward increases with policy complexity" Did the authors mean for the BA algorithm? This does not seem to be the case for SNIS.

13. line 335 (and throughout this paragraph): Fig A-B should be Fig 4A-B, similarly for Fig C-E.

14. line 349: "The first prediction arise" -> arises

**Have the authors made all data and (if applicable) computational code underlying the findings in their manuscript fully available?**

Reviewer #1: Yes

Reviewer #2: Yes

PLOS authors have the option to publish the peer review history of their article (what does this mean?). If published, this will include your full peer review and any attached files.

Reviewer #1: **Yes: **Tailia Malloy

Reviewer #2: No

**Figure resubmission:**
---

## [Decision Letter · Decision Letter 2]

18 Aug 2025

Dear Mr. Liu,

We are pleased to inform you that your manuscript 'Action subsampling supports policy compression in large action spaces' has been provisionally accepted for publication in PLOS Computational Biology.

Best regards,

Alex Leonidas Doumas

Academic Editor

PLOS Computational Biology

Andrea E. Martin

Section Editor

PLOS Computational Biology

Reviewer's Responses to Questions

**Comments to the Authors:**

Reviewer #1: I agree that the representation of the state-independent action distribution was my mistake. Thank you for clarifying that. This was my only comment from the last revision apart from noticing and looking through the comments of reviewer 2. I have done that again this time and to the best of my ability it seems that all of those comments were addressed successfully.

I agree with the author's position on keeping the SNIS model in the main paper, it's an important and related model that has recent applications, which are added in the new paragraph, and without this comparison the paper would be lacking. It also clarifies what the main contribution of the proposed model is and when it performs differently from related models.

The other major comment from reviewer 2 was regrading the statistical analysis and I agree with the tests on the relative strength of the correlation being significant.

Reviewer #2: I'm satisfied with the changes made by the authors to the manuscript. I think the clarifications are helpful and will improve the accessibility of the manuscript.

**Have the authors made all data and (if applicable) computational code underlying the findings in their manuscript fully available?**

Reviewer #1: Yes

Reviewer #2: Yes

PLOS authors have the option to publish the peer review history of their article (what does this mean?). If published, this will include your full peer review and any attached files.

Reviewer #1: **Yes: **Tailia Malloy

Reviewer #2: No

---

## [Editor Report · Acceptance letter]

PCOMPBIOL-D-24-02072R2

Action subsampling supports policy compression in large action spaces

Dear Dr Liu,

I am pleased to inform you that your manuscript has been formally accepted for publication in PLOS Computational Biology. Your manuscript is now with our production department and you will be notified of the publication date in due course.

With kind regards,

Judit Kozma
